# Combined lineage tracing and scRNA-seq reveals unexpected first heart field predominance of human iPSC differentiation

Francisco X Galdos[1,2], Carissa Lee[1], Soah Lee[3], Sharon Paige[1,4], William Goodyer[1,4], Sidra Xu[1], Tahmina Samad[1], Gabriela V Escobar[1], Adrija Darsha[5], Aimee Beck[1], Rasmus O Bak[6], Matthew H Porteus[1,7], Sean M Wu[1,2,8]*

[1]Stanford Cardiovascular Institute, Stanford University, Stanford, United States; [2]Institute for Stem Cell Biology and Regenerative Medicine, Stanford University, Stanford, United States; [3]Department of Pharmacy, Sungkyunkwan University, Stanford, United States; [4]Division of Pediatric Cardiology, Department of Pediatrics, Stanford University, Stanford, United States; [5]School of Medicine, University of California, San Diego, San Diego, United States; [6]Department of Biomedicine, Aarhus University, Aarhus C, Denmark; [7]Department of Pediatrics, Stanford University, Stanford, United States; [8]Division of Cardiovascular of Medicine, Department of Medicine, Stanford University, Stanford, United States

*For correspondence: smwu@stanford.edu

Competing interest: The authors declare that no competing interests exist.

**Abstract** During mammalian development, the left and right ventricles arise from early populations of cardiac progenitors known as the first and second heart fields, respectively. While these populations have been extensively studied in non-human model systems, their identification and study in vivo human tissues have been limited due to the ethical and technical limitations of accessing gastrulation-stage human embryos. Human-induced pluripotent stem cells (hiPSCs) present an exciting alternative for modeling early human embryogenesis due to their well-established ability to differentiate into all embryonic germ layers. Here, we describe the development of a TBX5/MYL2 lineage tracing reporter system that allows for the identification of FHF- progenitors and their descendants including left ventricular cardiomyocytes. Furthermore, using single-cell RNA sequencing (scRNA-seq) with oligonucleotide-based sample multiplexing, we extensively profiled differentiating hiPSCs across 12 timepoints in two independent iPSC lines. Surprisingly, our reporter system and scRNA-seq analysis revealed a predominance of FHF differentiation using the small molecule Wnt-based 2D differentiation protocol. We compared this data with existing murine and 3D cardiac organoid scRNA-seq data and confirmed the dominance of left ventricular cardiomyocytes (>90%) in our hiPSC-derived progeny. Together, our work provides the scientific community with a powerful new genetic lineage tracing approach as well as a single-cell transcriptomic atlas of hiPSCs undergoing cardiac differentiation.

## Editor's evaluation

This study presents elegant lineage tracing results demonstrating that first heart field (FHF) generates a dominance (>90%) of left ventricular cardiomyocytes in human iPSCs. The authors developed a TBX5/MYL2 reporter system in order to demonstrate this, and have supported their results utilizing single-cell RNA-sequencing with oligonucleotide-based sample multiplexing and this also provides a single-cell transcriptomic atlas of human iPSCs undergoing cardiac differentiation. These

differentiation pathways have been extensively studied in non-human models but this is the first demonstration of FHF progenitors giving rise to left ventricular cardiomyocytes in a human model system.

## Introduction

The human heart is one of the first organs to develop during embryogenesis with critical events in progenitor specification and differentiation occurring during the first 3 weeks of human gestation (*Buckingham et al., 2005*; *Cui et al., 2019*; *Hikspoors et al., 2022*; *Meilhac and Buckingham, 2018*; *Tan and Lewandowski, 2020*). Due to ethical and technical limitations in the study of human embryogenesis prior to 5 weeks gestation, developmental biologists have largely relied upon animal models to study cardiac development (*Hyun et al., 2021*; *Meilhac and Buckingham, 2018*). Early studies in mammalian cardiac progenitor biology identified the presence of two definitive multipotent progenitor populations known as the first (FHF) and second (SHF) heart fields which give rise to the left and right ventricles, respectively (*Cai et al., 2003*; *Dyer and Kirby, 2009*; *Meilhac et al., 2004*; *Mjaatvedt et al., 2001*; *Moretti et al., 2006*; *Waldo et al., 2001*). Furthermore, early lineage tracing studies using the mesodermal progenitor marker, *Mesp1*, has revealed that the early specification of these lineages likely occurs during the earliest stages of gastrulation, with the FHF emerging as the first wave of cardiac progenitors, followed by the SHF (*Lescroart et al., 2014*; *Saga et al., 1999*; *Scialdone et al., 2016*). With the advent of scRNA-seq, these progenitor populations have been extensively characterized and shown to exhibit unique transcriptional expression profiles (*de Soysa et al., 2019*; *Hill et al., 2019*; *Xiong et al., 2019*). Moreover, scRNA-seq profiling of murine left and right ventricles during early cardiac development has shown that transcriptional differences can be detected up to E10.5 of murine development, suggesting that early left and right ventricular development is characterized by unique transcriptional regulatory networks (*DeLaughter et al., 2016*; *Li et al., 2019*; *Li et al., 2016*).

While the distinct identities of the first and second heart field progenitors have been well established in the murine system, the identification of these progenitor populations within a human model has been severely limited by a lack of access to human embryonic tissues. Over the past decade, the advent of hiPSCs has allowed for the developmental modeling of multiple different embryonic lineages in vitro (*Holloway et al., 2020*; *Kanton et al., 2019*; *Karagiannis et al., 2019*; *Lian et al., 2013*; *Takahashi and Yamanaka, 2006*; *Yamanaka, 2008*). In the cardiac field, small molecule-based protocols modulating WNT signaling have become standard due to their remarkable efficiency in generating large numbers of beating cardiomyocytes that can be utilized for disease modeling, drug discovery, and the study of cellular functions (*Burridge et al., 2015*; *Chen et al., 2016*; *Feyen et al., 2020*; *Lian et al., 2013*; *Sacchetto et al., 2020*). Several questions remain as to whether hiPSC cardiac differentiations are capable of modeling early cardiac progenitor biology as seen during in vivo mouse development (*Protze et al., 2019*). Moreover, evidence is lacking as to whether current hiPSC differentiation protocols give rise to FHF- and SHF-derived LV and RV cardiomyocytes, respectively (*Protze et al., 2019*).

A major bottleneck in the identification of these cell types during hiPSC differentiation is the lack of lineage tracing tools that have been extensively used in murine models to understand the developmental lineage contributions of progenitor populations (*Barnes et al., 2010*; *Cai et al., 2003*; *Meilhac et al., 2004*; *Moretti et al., 2006*; *Tyser et al., 2020*; *Vincentz et al., 2017*; *Zhang et al., 2021*). Early studies profiling the expression of the T-box transcription factor, *TBX5*, identified its specific expression at the cardiac crescent and its role as a marker of early FHF progenitors (*Bruneau et al., 2001*; *Bruneau et al., 1999*). More recently, studies using inducible CreER/LoxP lineage tracing have shown the exquisite specificity of *TBX5* to label ventricular cardiomyocytes on the left but not the right, demonstrating a clear boundary between cell origins of left and right ventricular cardiomyocytes during embryogenesis (*Devine et al., 2014*). While lineage tracing tools have provided insight into the cellular contributions of the FHF in mice, no lineage tracing tool is currently available for tracing LV and RV cardiomyocytes in a human model system.

To identify FHF-progenitors and their derived cell types during hiPSC differentiation, we used a CRISPR-Cas9 targeting platform to engineer a TBX5 expression-driven, highly-sensitive, Cre/LoxP lineage tracing system in hiPSCs that contain a ventricular cardiomyocyte-specific myosin light chain-2

(*MYL2*)-tdTomato fluorescent protein. By conducting a time course analysis of cardiomyocyte differentiation, we identified a left ventricular cardiomyocyte predominant differentiation across two distinct cell lines based on the high percentage of TBX5-lineage positive ventricular cardiomyocytes (>90%). Using chemically modified lipid-oligonucleotides (CMOs), we conducted multiplexed scRNA-seq assays on 12 different timepoints across two independent hiPSC lines. Using differentiation trajectory analysis, we compared our scRNA-seq data with murine heart field development scRNA-seq data and validate the FHF origin and LV identity of cardiomyocytes generated. Finally, we conduct a comparison of our scRNA-seq data with a recently published 3D cardiac organoid differentiation (*Drakhlis et al., 2021*) and identify the greater potential of a 3D system to generate SHF-derived cell types. Together, our findings provide a powerful new tool for human in vitro cardiac development studies and a validated single-cell expression atlas for identifying the human FHF lineage during in vitro hiPSC differentiation.

## Results

### Generation of a TBX5-lineage tracing and ventricular reporter line by CRISPR/Cas9 genome editing

Given the well-established role of the T-box transcription factor, *Tbx5*, as a specific marker of the early FHF and left ventricular lineage (*Bruneau et al., 2001*; *Bruneau et al., 1999*; *DeLaughter et al., 2016*; *Devine et al., 2014*), we engineered a fluorescent lineage tracing system that would allow for the determination of whether *TBX5* lineage tracing could correctly identify left ventricular cardiomyocytes using a human iPSC model of cardiac differentiation. Previously, our laboratory developed an *MYL2*-tdTomato construct targeting a P2A-TdTomato to the stop codon of the *MYL2* gene that was validated to specifically isolate ventricular cardiomyocytes during hiPSC differentiations (*Chirikian et al., 2021*).To construct a reporter system that could isolate left ventricular cardiomyocytes, we employed a triple construct system that would allow for the identification of *MYL2*-positive ventricular cardiomyocytes and the identification of *TBX5*-lineage-positive left ventricular cardiomyocytes (*Figure 1A*). To lineage trace *TBX5* expressing cells during hiPSC differentiation, we developed two new genetic constructs based on a P2A self-cleaving peptide system that allows the tethering of genetic construct expression with a gene of interest (*Liu et al., 2017*). The first construct consists of tandem P2A-Cre Recombinase genes that are targeted to replace the stop codon of *TBX5* (*Figure 1B*). The second construct consists of a constitutively active CMV promoter followed by a floxed stop cassette and a downstream TurboGFP with the goal to only allow for TurboGFP expression after the excision of the stop cassette by Cre (*Figure 1B*). Using CRISPR/Cas9 genome editing, we first targeted the MYL2-tdTomato construct into two hiPSC lines derived from healthy donor patients (*Figure 1B*). Using an inside-out PCR strategy (*Galdos et al., 2021*; *Ran et al., 2013*), we confirmed the successful integration of the MYL2 construct based on the integration of the 5' and 3' ends of the construct and selected a heterozygous integrated clone (*Figure 1C*). We subsequently integrated the CMV-Lox-STOP-Lox-TurboGFP construct into the CCR5 safe harbor site and confirmed successful integration via inside-out PCR (*Figure 1C*). Next, we integrated the P2A-Cre-P2A-Cre construct into the *TBX5* locus by replacing the stop codon of the gene. To ensure maximal sensitivity of our lineage tracing system, we integrated the *P2A-Cre* construct in a homozygous manner to ensure high expression of Cre recombinase upon expression of TBX5 (*Figure 1C*). Importantly, the expression of *TBX5* is preserved with this approach since the Cre recombinase is inserted after the *TBX5* coding sequence and the fusion protein product undergoes self-cleaving at the P2A sequence (*Liu et al., 2017*). Using sanger sequencing, we validated the in-frame integration of the P2A sequences of both the *MYL2* and *TBX5* constructs (*Figure 1D*). Lastly, we confirmed the maintenance of pluripotency after three rounds of genome editing by immunostaining of pluripotency marker *OCT4, NANOG, and TRA-1-8-1* (*Figure 1E*), thus demonstrating the successful genome editing of three independent genetic constructs into two different hiPSC lines.

### TBX5-lineage/MYL2 reporter system reveals predominance of left ventricular differentiation using small molecule WNT protocol

To determine the proportion of hiPSC-derived cardiomyocytes that exhibit a TBX5-lineage positive phenotype, we conducted cardiac differentiations using a widely published differentiation protocol

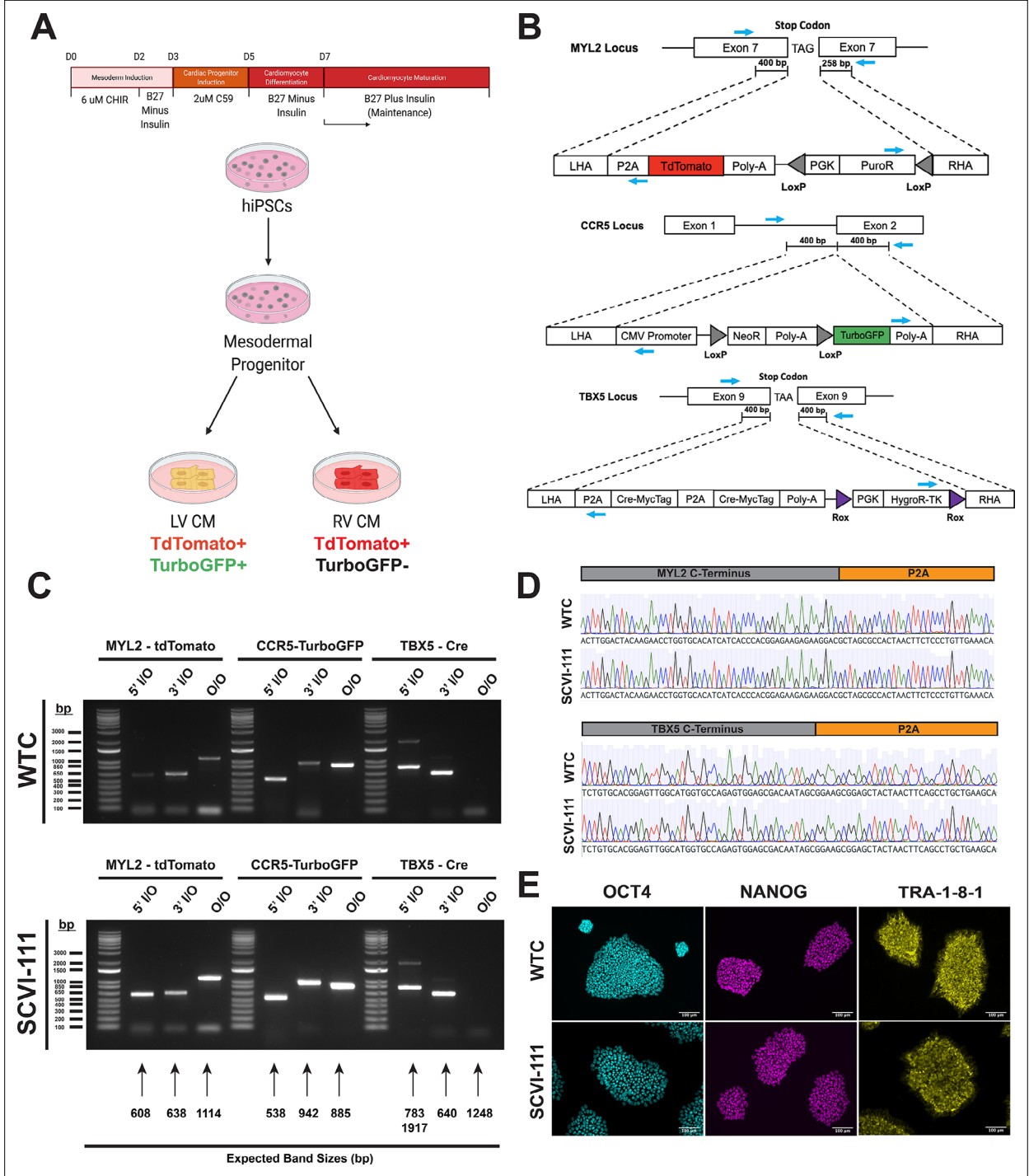

**Figure 1.** Integration of T-box transcription factor (TBX5)/myosin light chain-2 (MYL2) lineage tracing reporter system into human-induced pluripotent stem cells. (**A**) Schematic of lineage tracing strategy for identifying left ventricular cardiomyocytes in vitro. (**B**) CRISPR/Cas9 gene targeting strategy of genetic constructs for TBX5 lineage tracing and MYL2 direct reporter. MYL2, CCR5, and TBX5 constructs contain Puromycin (PuroR), Neomycin (NeoR), and Hygromycin (HygroR) resistance cassettes for the selection of human-induced pluripotent stem cell (hiPSC) after targeting of genetic constructs. Blue arrows indicate the location of PCR primer binding sites for confirmation of construct integration. LHA = Left Homology Arm, RHA = Right Homology Arm. (**C**) Inside-Outside (I/O) and Outside-Outside (O/O) PCR DNA agarose gels for confirmation of integration of genetic constructs into MYL2, CCR5, and TBX5 genetic loci. Inside, represents a primer inside the construct region while Outside represents a primer that binds outside the homology arm regions of genetic constructs. Expected band sizes are noted with arrows for each lane. (**D**) Sanger sequencing traces for C-terminal regions of MYL2 and TBX5 genes indicating in-frame integration of P2A site. (**E**) Bright field and immunofluorescence images of pluripotency marker expression in hiPSC lines after integration of all three genetic constructs.

*Figure 1 continued on next page*

*Figure 1 continued*

The online version of this article includes the following source data and figure supplement(s) for figure 1:

**Source data 1.** Raw and uncropped DNA electrophoresis image data for genotyping PCR for genetic constructs presented in *Figure 1C*.

**Source data 2.** Single guide RNA sequences used for gene targeting.

**Source data 3.** Genotyping primer sequences for identification of genetic constructs.

**Figure supplement 1.** Flow cytometry analysis of TurboGFP expression in genome-edited human-induced pluripotent stem cells (hiPSCs) after 30 days of pluripotency culture.

consisting of biphasic activation and subsequent inhibition of WNT signaling using small molecules (*Figure 1A*, **see Methods**; *Lian et al., 2013*). We employed a strategy where we conducted a high throughput flow cytometry analysis of cardiac troponin, TurboGFP, and tdTomato expression across multiple timepoints during cardiac differentiation and across the two independent cell lines containing our reporter system (*Figure 2A, B and C*). Analysis of *TNNT2* expression from day 3 to 30 of differentiation revealed a gradual upregulation of *TNNT2* expression starting at day 7 of differentiation (*Figure 2B and D*), with the greatest increase in *TNNT2+* cardiomyocytes being reported between day 7 and 11 of differentiation. Overall cardiac differentiation at day 30 across both reporter lines (WTC and SCVI-111) averaged 93.2 ± 0.80% and 92.1 ± 1.60% of *TNNT2+* cells out of the total cells analyzed, respectively (*Figure 2D*). We further analyzed the proportion of cells that were positive for TurboGFP + between day 3 and 30 of differentiation (*Figure 2B and E*) and found a large increase in TurboGFP + cells between days 7 and 11 with a continuous increase in the level of GFP signal as the differentiation proceeded. By day 30, both WTC and SCVI-111 lines exhibited 99.0 ± 0.21% and 93.2 ± 0.84% of total TNNT2 + cardiomyocytes expressing TurboGFP, respectively, indicating a predominance of cardiomyocytes from the TBX5 lineage (*Figure 2E*).

Since TBX5 is known to be expressed in both atrial and ventricular cardiomyocytes, we next determined the percentage of ventricular cardiomyocytes that are within the TBX5-lineage by analyzing the proportion of MYL2-tdTomato+ cardiomyocytes that express TurboGFP. *MYL2* expression gradually increases over time during both hiPSC cardiac differentiation and in vivo development and is highly tied to the overall maturational status of hiPSC-derived cardiomyocytes (*Bizy et al., 2013*; *Chirikian et al., 2021*; *DeLaughter et al., 2016*; *Li et al., 2016*; *O'Brien et al., 1993*). Consistent with previous studies, we show that the percentage of MYL2-tdTomato+ cardiomyocytes increases between days 15 and 30 of cardiomyocyte differentiation with some line-to-line variability likely tied to variation in hiPSC-CM maturation rate thus accounting for the higher percentage seen in the WTC line over the SCVI-111 (*Figure 2C and F*). Across day 15, 20, and 30 we observed that for both cell lines, the proportion of ventricular cardiomyocytes marked by the tdTomato reporter were more than 95% for TurboGFP indicating that nearly all ventricular cardiomyocytes were within the TBX5-lineage (*Figure 2F*, *Figure 2G*, *Figure 3A*).

We further validated the expression kinetics of our reporter system by conducting bulk gene expression analyses using RT-qPCR across multiple timepoints during hiPSC differentiation (*Figure 3B–E*). We evaluated the expression of known markers of early FHF progenitors and left ventricular cardiomyocytes, *HAND1* and *TBX5* (*Barnes et al., 2010*; *de Soysa et al., 2019*; *Devine et al., 2014*; *Vincentz et al., 2017*). We also evaluated the expression of Cre recombinase throughout differentiation. Relative to day 0 we observed that all three markers exhibited high expression values with *HAND1* exhibiting more than 50,000 fold upregulation relative to day 0 by day 7 of differentiation across both lines (*Figure 3B*). Similarly, by day 30 of differentiation, *TBX5* exhibited nearly 3000-fold upregulation relative to day 0 (*Figure 3B*). The expression of Cre recombinase increased over time and was consistent with the expected increase in TurboGFP expression observed in the flow cytometry data (*Figure 3B*).

In addition to analyzing FHF marker expression, we also examined the expression of SHF markers such as *ISL1*, *FGF8*, and *TBX1* (*Figure 3C*; *Cai et al., 2003*; *Park et al., 2008*; *Rana et al., 2014*). While *ISL1* has been reported to be expressed in the early FHF lineage (*Ma et al., 2008*), a well-established observation is that *ISL1* expression is sustained during the emergence of the SHF (*Cai et al., 2003*). Interestingly, we observed in both the WTC and SCVI-111 that *ISL1* expression peaked at day 5 of differentiation, which is indicative of an early cardiac progenitor population at that timepoint. We did not observe a sustained expression of *ISL1* and rather observed its downregulation over time. Similarly, *FGF8* has been reported to be important for early cardiomyocyte differentiation

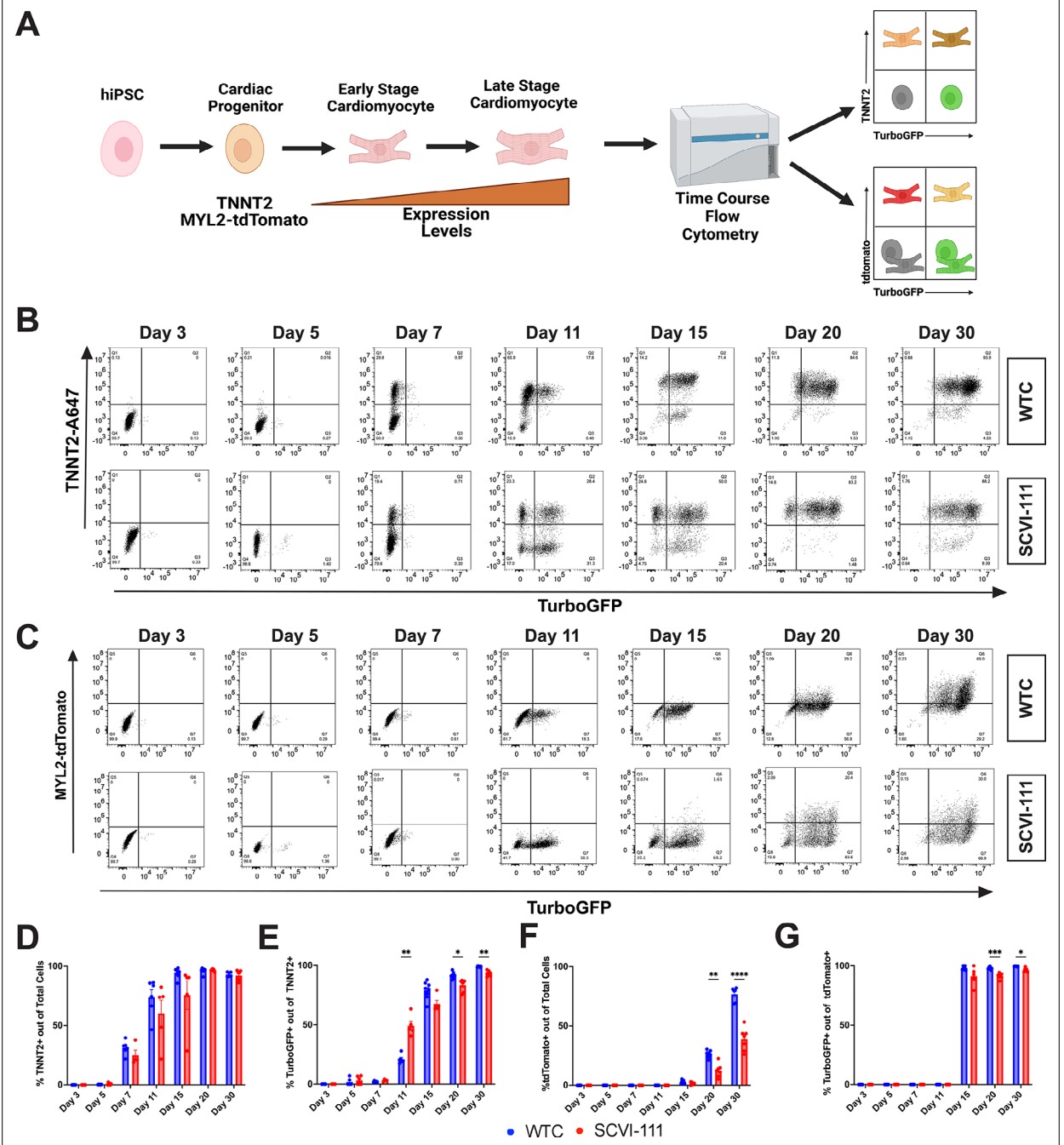

**Figure 2.** T-box transcription factor (TBX5)-lineage tracing reveals a predominance of lineage positive cardiomyocytes through the course of human-induced pluripotent stem cell (hiPSC) cardiac differentiation. (**A**) Schematic of analysis approach of reporter system expression through the course of cardiac differentiation. (**B**) Representative flow cytometry plots for the expression of TurboGFP and cardiac troponin T staining between day 3–30 of differentiation for WTC and SCVI-111 reporter lines. Gating set based on day 3 of differentiation. (**C**) Representative flow cytometry for the expression MYL2-tdTomato and TurboGFP between day 3 and30. Gating set based on day 3 of differentiation. (**D**) Quantification of percentage cardiac troponin T (TNNT2) expressing cells across all cells sampled. (**E**) Quantification of the percentage of TurboGFP-positive cells out of cardiac troponin-positive cardiomyocytes. (**F**) Quantification of tdTomato expressing cells across total cells sampled. Quantification of TurboGFP-positive cells out of MYL2-Tdtomato cardiomyocytes. N=4-8 independent biological replicates were collected per sample. Statistical significance determined by two-way ANOVA with first independent variable analyzed being time and the second variable being cell line. *p <0.05, **p<0.01, ***p<0.001, ****p<0.0001. Error bars represent standard error of the mean (SEM).

The online version of this article includes the following figure supplement(s) for figure 2:

**Figure supplement 1.** FACS sorted TurboGFP positive cells enrich for T-box transcription factor (TBX5) xxpression.

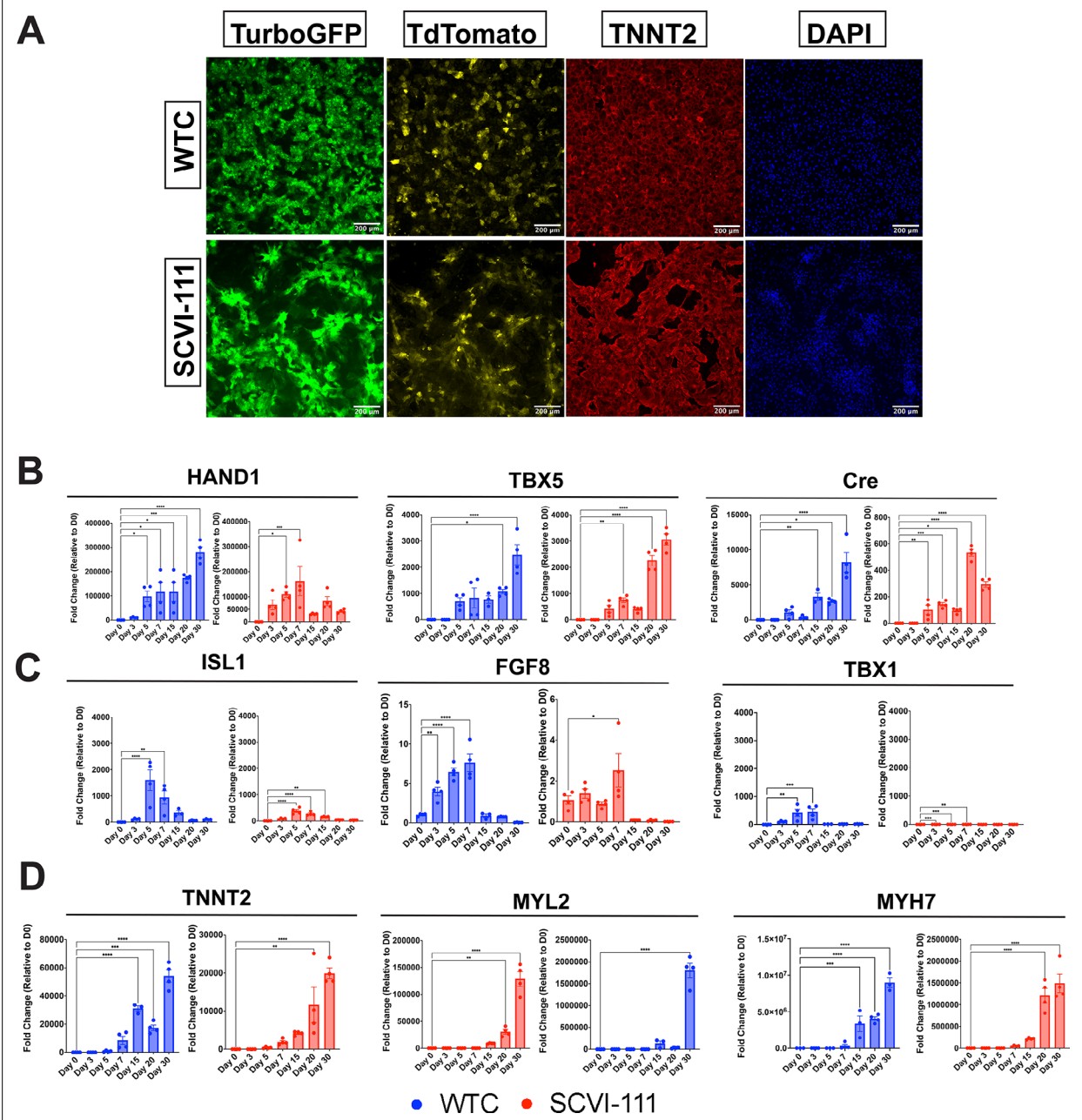

**Figure 3.** Immunofluorescence imaging of day 30 reporter cardiomyocytes and RT-qPCR profiling of first heart field (FHF) and second heart field (SHF) markers across differentiation. (**A**) Immunofluorescence images of TurboGFP, TdTomato, and TNNT2 expression at day 30 of cardiac differentiation for WTC and SCVI-111 reporter lines. (**B**) RT-qPCR profiling of FHF markers HAND1 and TBX5 along with Cre Recombinase between day 0 and 30 of differentiation. (**C**) RT-qPCR profiling of pan-cardiac progenitor marker ISL1, and SHF markers TBX1 and FGF8 between day 030 of differentiation. (**D**) RT-qPCR profiling of pan-cardiomyocyte marker TNNT2. (**E**) RT-qPCR profiling of ventricular marker myosin light chain-2 (MYL2). N=3–4 biological replicates per timepoint. Significance determined by one-way ANOVA with Dunnet's multiple comparisons correction. *p<0.05, **p<0.01, ***p<0.001, ****p<0.0001. Error bars represent standard error of the mean (SEM).

The online version of this article includes the following source data for figure 3:

**Source data 1.** qPCR human primer sequences.

in both the FHF and SHF, however, its expression is maintained within SHF progenitors during cardiogenesis. We observed that while *FGF8* expression was present at the mesodermal stage of differentiation (*Figure 3C*), a significant drop in expression was observed after day 7. *TBX1*, a pharyngeal endoderm and mesoderm marker (*Chapman et al., 1996*; *Mesbah et al., 2012*; *Rana et al., 2014*;

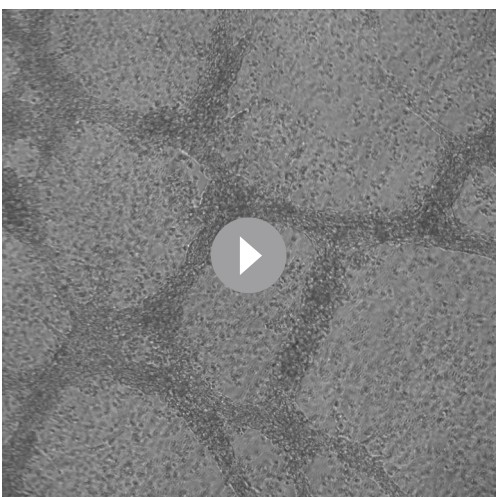

**Video 1.** Contractility of SCVI-111 reporter line cardiomyocytes at day 15 of differentiation.
https://elifesciences.org/articles/80075/figures#video1

*Vitelli et al., 2002*), also known as a marker of the anterior second heart field (*Liao et al., 2008*; *Meilhac and Buckingham, 2018*; *Nevis et al., 2013*), was unexpectedly upregulated during gastrulation and early cardiac progenitor stages of differentiation but declined after day 7 of differentiation with some line-to-line and batch-to-batch variability (*Figure 3C*). Importantly, *TBX1* was not upregulated to the same degree as either *HAND1* or *TBX5, both of* which exhibited greater than 1000-fold upregulation in both lines analyzed. This lack of SHF markers upregulation after day 7 contrasts with the continued or even increased expression of FHF markers *HAND1* and *TBX5*, thus indicating the predominance of FHF-derived cells at later stages of differentiation. Notably, the fold change expression of TBX1 and ISL1 was significantly lower than that of *HAND1 and TBX5,* with the FHF markers exhibiting more than 100,000- and 1000-fold increases, respectively (*Figure 2B and C*). Lastly, we also observed an increase in cardiomyocyte maturation markers such as *TNNT2, MYL2, and MYH7* (*Figure 3E*) indicating the reporter lines fully differentiated into beating cardiomyocytes (*Videos 1 and 2*). Consistent with our flow cytometry data, we also observe a gradual upregulation of *MYL2* through the course of hiPSC cardiac differentiation. (*Figure 2B and E*).

## scRNA-seq time course reveals three major developmental trajectories during hiPSC differentiation

Given that our Cre/LoxP-based fluorescent reporter system showed a predominance of *TBX5*-lineage cardiomyocytes (*Figure 2E*) and our qPCR data showed an upregulation of FHF, but not SHF, gene markers at late stages of differentiation (*Figure 3B and C*), we asked whether scRNA-seq may help to pinpoint the developmental trajectories that bifurcate between FHF and SHF during hiPSC differentiation. Using sample multiplexing with CMO in our scRNA-seq experiment where we captured cells from both the WTC and SCVI-111 lines and at 12 different timepoints hiPSC cardiac differentiation (Days 0–7 and 11, 13, 15, and 30) for a total of 27,595 cells after sample demultiplexing and quality control (*Figure 4A*, *Figure 4—figure supplements 1–3*).

Using a well-published batch correction method known as the mutual nearest neighbor algorithm (*Haghverdi et al., 2018*), we batch-corrected the effect of the scRNA-seq runs and conducted downstream dimensionality reduction and unsupervised clustering (*Figure 4—figure supplement 4*). Annotation of unsupervised clusters revealed 13 major populations during cardiac differentiation (*Figure 4* and *Figure 4—figure supplement 4*, *Figure 4—source data 2*). During the early days of differentiation we identified cell populations consistent with pluripotent stem cells, primitive streak, and definitive endoderm populations marked by the expression of *POU5F1, MIXL1,* and *SOX17*, respectively (*Figure 4—figure supplement 4*, *Figure 4—source data 2*; *Mead et al., 1996*; *Pijuan-Sala et al., 2019*; *Takahashi and Yamanaka, 2006*; *Tyser et al., 2021*). By day 3 of differentiation, we

**Video 2.** Contractility of WTC reporter line cardiomyocytes at day 15 of differentiation.
https://elifesciences.org/articles/80075/figures#video2

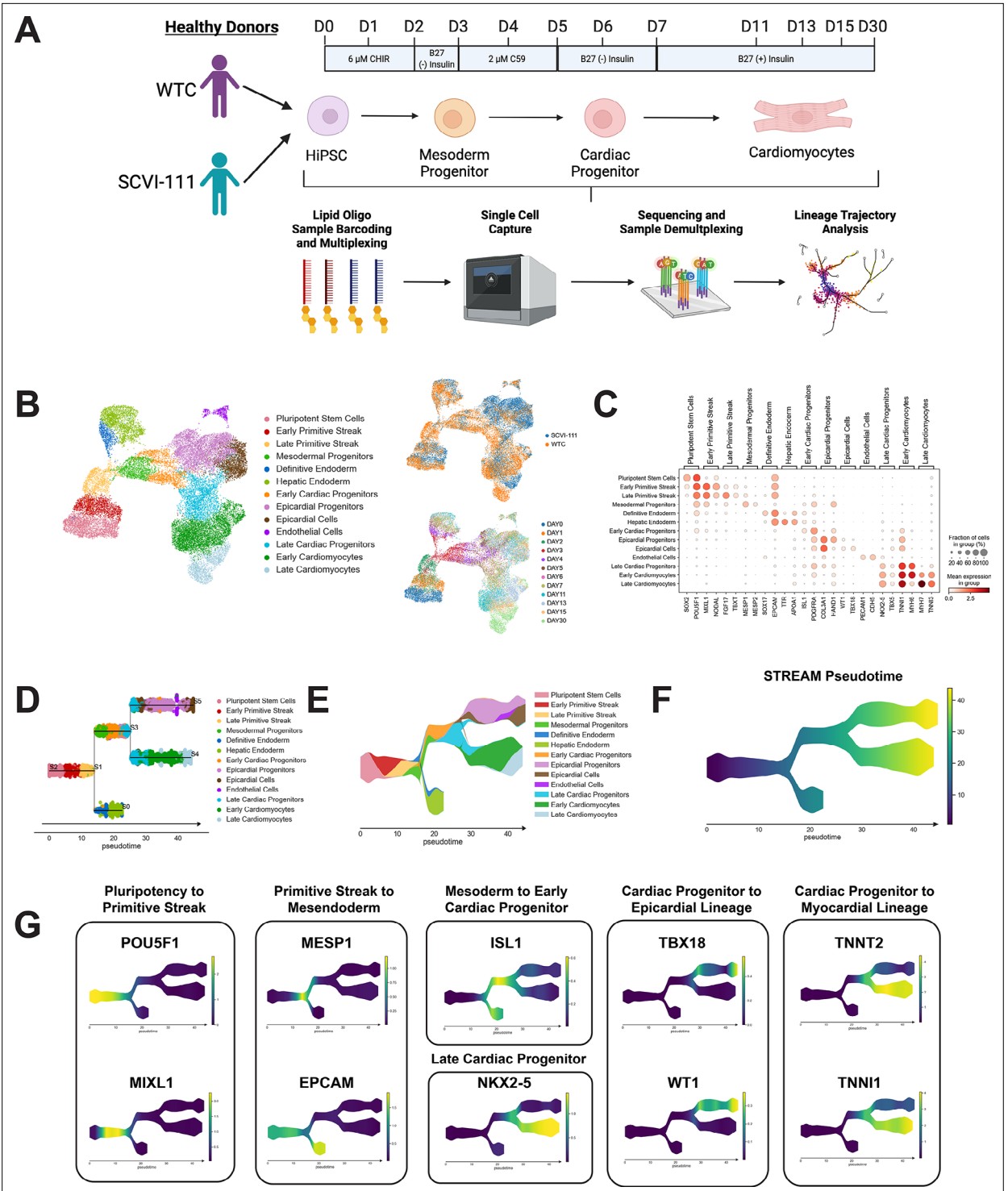

**Figure 4.** single-cell RNA sequencing (scRNA-seq) profiling and trajectory inference reveals emergence of myocardial and epicardial lineages during human-induced pluripotent stem cell (hiPSC) cardiac differentiation. (**A**) Diagram of scRNA-seq multiplexing for profiling of 12 timepoints across two lines during hiPSC cardiac differentiation. (**B**) Left, UMAP plot with identification of 13 cell populations over the course of cardiac differentiation. Right-Top, Plot indicating labeling of cells captured from WTC and SCVI-111 line. Right-Bottom, Plot indicating timepoints of differentiation at which cells were captured for scRNA-seq. N=27,595 single cells. (**C**) Dotplot presenting the expression of top markers for major cell populations identified during hiPSC cardiac differentiation. (**D**) Subway map plot showing the projection of cells along cell lineages detected using STREAM analysis. (**E**) Stream plot indicating the relative cell type composition along each branch of differentiation identified by STREAM. (**F**) Graph indicating pseudotime values

*Figure 4 continued on next page*

*Figure 4 continued*

calculated by STREAM for ordering cells along a continuous developmental projection axis. (**G**) Feature plots for top gene markers identified during each major developmental phase of hiPSC cardiac differentiation.

The online version of this article includes the following source data and figure supplement(s) for figure 4:

**Source data 1.** GEO Accession Numbers for Datasets.

**Source data 2.** Differentially Expressed Genes Identified For Annotated Cell Types.

**Source data 3.** Genes Correlated with Differentiation Trajectories Identified During hiPSC Cardiac Differentiation.

**Source data 4.** Description of scRNA-seq Run and hashtag oligo sequences used for sample multiplexing.

**Figure supplement 1.** Quality control and hashtag oligo labeling of samples for scRNA-seq sample Galdos_Seq_Run1.

**Figure supplement 2.** Quality control and hashtag oligo labeling of samples for scRNA-seq sample Galdos_Seq_Run2.

**Figure supplement 3.** Quality control and hashtag oligo labeling of samples for scRNA-seq sample Galdos_Seq_Run3.

**Figure supplement 4.** Unsupervised clustering and marker expression of combined human-induced pluripotent stem cell (hiPSC) single-cell RNA sequencing (scRNA-seq) data from WTC and SCVI-111 lines.

**Figure supplement 5.** Trajectory inference analysis of human-induced pluripotent stem cell (hiPSC) cardiac differentiation across WTC and SCVI-111 Lines.

**Figure supplement 6.** Feature plots of selected first heart field (FHF), second heart field (SHF), endoderm, and cardiomyocyte markers.

**Figure supplement 7.** Comparison of expression of atrial and ventricular cardiomyocyte markers in human-induced pluripotent stem cell (hiPSC)-CM single-cell RNA sequencing (scRNA-seq) time course.

observed the emergence of mesodermal progenitors, early cardiac progenitors, late cardiac progenitors, cardiomyocytes, and epicardial populations marked by the expression of *MESP1, ISL1, NKX2-5, TNNT2*, and *WT1*, respectively (*Figure 4—figure supplement 4*, *Figure 4—source data 2*; *Barnes et al., 2010*; *Christoffels et al., 2009*; *Rudat and Kispert, 2012*; *Zeng et al., 2011*).

Given the identification of distinct cell types within our single-cell data, we asked whether we could further identify developmental trajectories during hiPSC cardiac differentiation. We used a Python-based bioinformatic pipeline known as STREAM to automatically identify and visualize differentiation trajectories within our scRNA-seq data (*Chen et al., 2019*). STREAM uses a low-dimensional manifold such as a UMAP plot and calculates a principal graph that identifies differentiation paths throughout the dataset. Intriguingly, the STREAM algorithm fits a principal graph that identified two major bifurcations during hiPSC differentiation (*Figure 4D* and *Figure 4—figure supplement 5*). Along with the fitting of a principal graph, we calculated STREAM pseudotime by setting the pluripotent stem cell cluster as the root of the differentiation. We then reordered and projected cells along the principal graph according to increasing pseudotime to visualize our annotated cell types as they progressed during differentiation (*Figure 4D–F* and *Figure 4—figure supplement 5*). To further characterize the distinct trajectories identified by STREAM, we correlated the expression of gene expression with the cell pseudotime along each unique branch (*Figure 4— source data 3*). This analysis recovered multiple gene markers known to be expressed during the course of gastrulation and cardiac development. The first bifurcation identified occurred at the late primitive streak stage and represents the bifurcation into mesodermal and endodermal cell lineages. Gene markers identified for the endodermal lineage include markers such as *EPCAM and FOXA2*, as well as hepatic-like endodermal markers such as *APOA1* and *AFP* (*Figure 4G*; *Hurrell et al., 2019*; *Pijuan-Sala et al., 2019*; *Sarrach et al., 2018*). Along the mesodermal differentiation path, we observed the upregulation of *MESP1* followed by ISL1 expression as cells became specified along the cardiac progenitor lineage. ISL1 expression preceded the expression of *NKX2-5* and encompasses both endodermal precursors as well as cardiac progenitors, supporting the earlier but less cardiac-specific expression of *ISL1*. At the second bifurcation, we identified an *NKX2-5* population that bifurcated into myocardial and epicardial lineages. Top ranking markers for the epicardial lineage included known epicardial markers *TBX18, WT1, TCF21*, and *IGF2*, while the myocardial lineage was characterized by the elevated expression of sarcomeric genes (*Figure 4G* and *Figure 4—figure supplement 5D*; *Christoffels et al., 2009*; *Hu et al., 2020*; *Li et al., 2011*; *Rudat and Kispert, 2012*; *Tandon et al., 2013*). Within the epicardial lineage, we observed a high expression of *HAND1* along with the enrichment of extracellular matrix genes such as *COL3A1, VIM*, and *COL1A1* at the epicardial progenitor population, suggesting the emergence of *WT1* positive

epicardial cells from a *HAND1* expressing precursor (*Figure 4—source data 3*). Together, STREAM revealed the emergence of an epicardial and myocardial lineage from a common cardiac progenitor during hiPSC differentiation.

## Predominance of FHF cardiomyocyte differentiation by hiPSCs confirmed by comparison with scRNA-seq data from murine heart field development

To confirm the FHF cardiomyocyte-predominant differentiation of hiPSCs that we observed in our TBX5 lineage tracing (*Figure 2D*) and qPCR (*Figure 3B and C*) data, we conducted a comparison between previously published murine scRNA-seq heart field data (*de Soysa et al., 2019*; *Hill et al., 2019*; *Pijuan-Sala et al., 2019*) and our hiPSC cardiac differentiation data. We clustered data from the murine datasets representing seven major cell types of interest including nascent mesoderm, heart field progenitors, epicardial cells, left and right ventricular cardiomyocytes, and outflow tract cardio-myocytes (*Figure 5A*). As previously reported (*de Soysa et al., 2019*; *Hill et al., 2019*), we observed a bifurcation of the FHF and SHF cells from the nascent mesoderm and observed a clear contribution of both heart field progenitors to the development of left and right ventricular cardiomyocytes with FHF cells contributing to the LV and SHF cells contributing to the RV/OFT. Intriguingly, we observed the epicardial lineage branched off from FHF progenitor cells (*Figure 5B*). This observation is consistent with the recent lineage tracing literature that indicates the contribution of a subset of FHF progenitors to both left ventricular cardiomyocytes as well as epicardial cells (*Tyser et al., 2020*; *Zhang et al., 2021*).

To further dissect the gene expression changes that occur during FHF and SHF development, we replotted each FHF and SHF cell population using STREAM and displayed the expression of known FHF and SHF progenitors markers during murine heart field development and hiPSC cardiac differentiation (*Figure 5B*). Consistent with the literature, *TBX5* and *HCN4* were upregulated in FHF cells during mouse development in vivo and are completely absent in the aSHF lineage (*Figure 5C*; *Andersen et al., 2018*; *Bruneau et al., 2001*; *Devine et al., 2014*; *Später et al., 2013*). Of note, *Tbx5* appeared to gradually increase in expression during the transition from FHF progenitors to LV CMs with a gradual downregulation as development progresses, indicating a dynamic expression pattern through the course of development. In contrast to the FHF markers, we observed a clear upregula-tion of aSHF markers *TBX1* and *FGF8* during early aSHF progenitor development in mice with *Fgf8* exhibiting its highest expression pattern at prior to the bifurcation between OFT CM and RV CM (*Figure 5C*; *Nevis et al., 2013*; *Park et al., 2008*; *Vitelli et al., 2002*). As expected, the expression of *TBX1* and *FGF8* was absent in FHF progenitors, albeit a low *Fgf8* expression was found in early LV CM, which is expected given the role of Fgf signaling during early cardiomyocyte differentiation (*Khosravi et al., 2021*; *Reifers et al., 2000*). We next plotted the expression of these markers during our hiPSC cardiac differentiations and observed a striking consistency in marker expression with the FHF lineage in the mouse (*Figure 5D*). Importantly, we observed the upregulation of TBX5 starting at the late progenitor stage and increasing during the myocardial branch of the differentiation, like the kinetics observed in the FHF trajectory of the mouse (*Figure 5D*). Similarly, *HCN4* expression remained high during cardiomyocyte differentiation further supporting the left ventricular identity of the myocardial branch given the reported role of HCN4 as an early FHF and LV marker (*Später et al., 2013*).

Having observed the similarities between murine FHF development and our hiPSC differentiations, we asked whether our hiPSC differentiations exhibited a ventricular-specific differentiation trajectory. We confirmed the ventricular-specific trajectory of our hiPSC differentiation given the gradual upreg-ulation of the ventricular-specific Iroquois transcription factor, *IRX4* (*Figure 5E*; *Nelson et al., 2016*; *Nelson et al., 2014*). Previous studies have shown *IRX4* to mark early ventricular-specific cardiomy-ocytes, which we effectively observed in both the left and right ventricular differentiation lineages in the murine data (*Nelson et al., 2014*). We further confirmed the ventricular-specific differentiation of our hiPSC-derived CMs by observing the absence of atrial markers, *KCNA5*, *NR2F1*, and *VSNL1* (*Figure 4—figure supplement 7D*). Lastly, consistent with the MYL2-tdtomato expression pattern observed with our reporter lines (*Figure 2F*), we observed the gradual upregulation of MYL2 in both the human and murine datasets. Together, our analysis provides evidence for the predominance of FHF ventricular cardiomyocyte development during hiPSC differentiation.

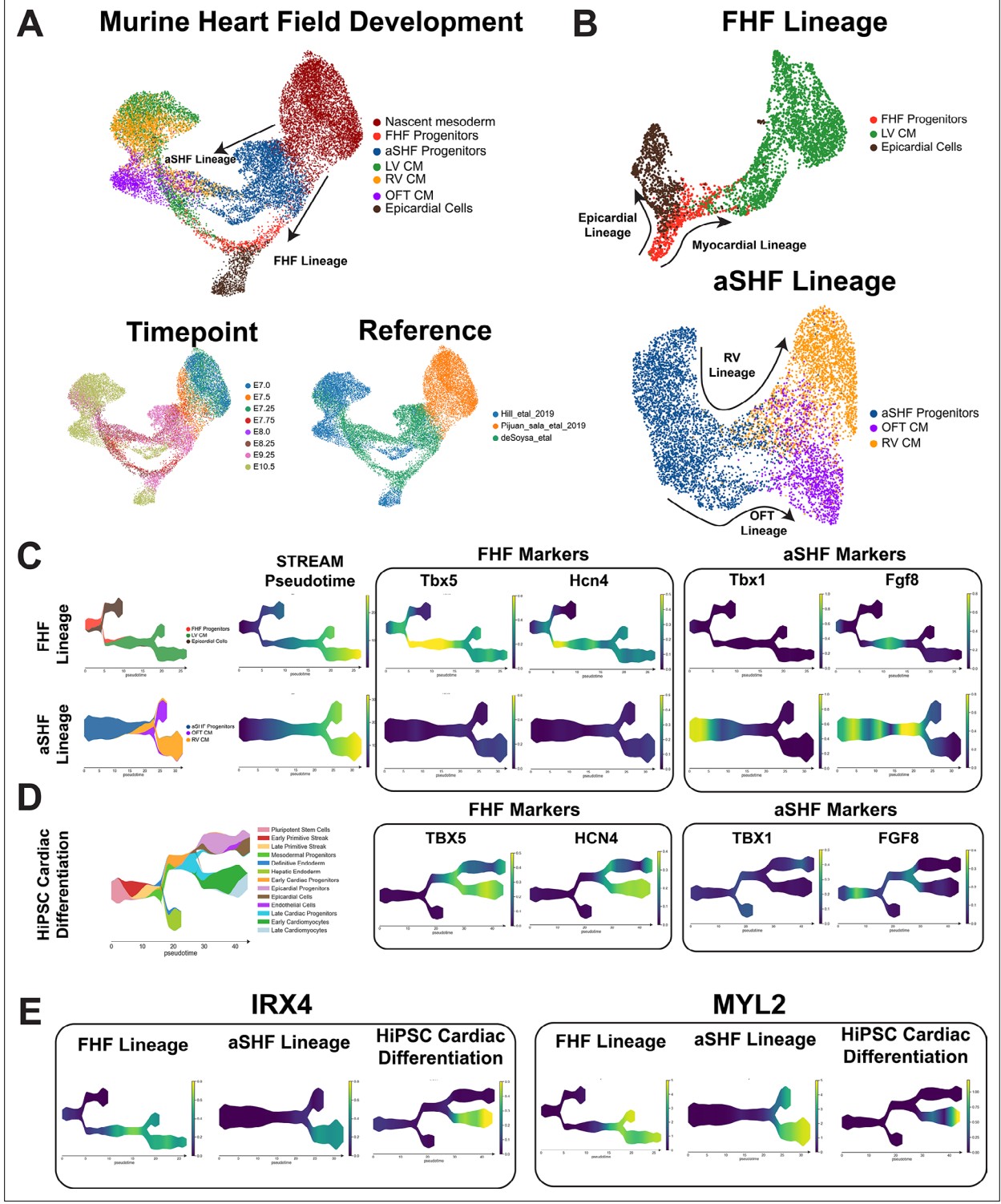

**Figure 5.** Comparison of heart field development between murine and human-induced pluripotent stem cell (hiPSC) cardiac differentiation reveals first heart field (FHF) identity of hiPSC cardiac lineages. (**A**) Top, UMAP plot showing clustering of murine cell types encompassing early mesodermal progenitors, FHF/second heart field (SHF) progenitors, LV/RV/OFT cardiomyocytes, and epicardial cells. Bottom-left, labeling of cell types by timepoint analyzed. Bottom-right, annotation by reference from which data was obtained. (**B**) UMAP embeddings of heart field development split by FHF and SHF lineages. (**C**) FHF and aSHF trajectory analysis and plotting of expression FHF (TBX5 and HCN4) and anterior SHF (TBX1 and FGF8) markers. (**D**) Analysis of FHF and aSHF marker expression during hiPSC cardiac differentiation. (**E**) Gene expression analysis of ventricular markers Iroquois transcription factor (IRX4) and myosin light chain-2 (MYL2) during hiPSC cardiac differentiation.

## Comparison of 2D and 3D cardiac differentiation uncovers potential of organoid system for SHF generation

Recently, multiple groups have proposed the use of 3D differentiation to better model chamber morphogenesis and potentially model both first and second heart field development in vitro (*Andersen et al., 2018*; *Drakhlis et al., 2021*; *Protze et al., 2019*; *Rossi et al., 2021*). Interestingly, murine gastruloid and precardiac organoids have been shown to exhibit aspects of first and second heart field development (*Andersen et al., 2018*; *Rossi et al., 2021*); however, data on the ability to generate both heart fields in human iPSCs are lacking. Given the predominance of FHF progenitors and LV cardiomyocytes made from our 2D hiPSC differentiation platform, we assessed whether a greater repertoire of cardiac cells can be generated from a 3D hiPSC differentiation platform by analyzing scRNA-seq data from a recently published cardiac organoid study (*Drakhlis et al., 2021*). This paper demonstrated the close relationship between anterior endoderm lineages and anterior second heart field cells during hiPSC differentiation by showing that anterior foregut endoderm can be generated alongside cardiac lineage cell types (*Kelly et al., 2001*; *Rochais et al., 2009*). We compared scRNA-seq data from our hiPSC-derived cardiac cells with those generated by the Drakhlis et al., group using their 3D differentiation protocol (*Drakhlis et al., 2021*). We first conducted a cross-dataset comparison of FHF and SHF marker analysis where we focused exclusively on cell types composing the myocardial lineages of both datasets (*Figure 6A*). In our 2D cardiac differentiation, we observed a clear increased expression of FHF markers *TBX5, HCN4, and HAND1* during differentiation with the absence of SHF markers *FGF8* and *TBX1* (*Figure 6B and C*). Interestingly, the day 13 data from Drakhlis et al., showed two clusters exhibiting distinct transcriptional expression patterns suggestive of FHF and SHF progenitors (*Figure 6D*), with both clusters appearing to give rise to *TNNI1* and *NKX2-5* positive cardiomyocytes. Cluster 2 and 8 of the cardiac organoid data indicated a high expression of *TBX5, HAND1*, along with the upregulation of *HCN4* during differentiation which was consistent with the FHF trajectory we observed in our 2D data (*Figure 6E*). Interestingly, we found that cells in clusters 4, 7, and 10 of Drakhlis et al., exhibited a high expression of *TBX1, FGF8*, and *ISL1*, all of which were consistent with a SHF identity (*Cai et al., 2003*; *Mesbah et al., 2012*; *Park et al., 2008*). Moreover, we also observed a cardiomyocyte population emerging from the *TBX1*+ population that was negative for FHF markers *TBX5* and *HCN4*, but highly expressing *ISL1* and *HAND1*, which is suggestive of an OFT CM population that is known to express *HAND1* and emerge from *ISL1* expressing SHF progenitors.

To determine whether this TBX5-ISL1+HAND1+ cardiomyocyte population was indeed OFT CMs, we conducted a joint analysis of the cardiomyocytes from our 2D differentiations and the Drakhlis et al., cardiac organoid. Using unsupervised clustering, we observed a cardiomyocyte population emerge from this study that displayed the absence of TBX5. Interestingly, we observed that the rest of the organoid cardiomyocytes co-clustered with the cells from our 2D differentiations indicating shared gene expression profiles (*Figure 6F*). We conducted differential gene expression analysis of the putative OFT CM cluster and the rest cardiomyocytes and found a statistically significant enrichment of markers associated with OFT development including *HAND1, BMP2, WNT5A*, and *PITX2* (*Délot et al., 2003*; *Li et al., 2016*; *Ma et al., 2013*; *Schleiffarth et al., 2007*), while the rest of the cardiomyocytes exhibited high expression of markers associated with early LV development such as *TBX5* and *NPPA* (*Figure 6G*, *Figure 6—source data 1*; *Li et al., 2016*). Overall, these data thus provide strong evidence for the emergence of a SHF-derived cell type within a 3D organoid differentiation protocol and reinforce the left ventricular identity that we identified using a standard 2D small molecule differentiation protocol.

## Discussion

Over the past decade, the development of highly efficient cardiac-directed differentiation protocols have significantly advanced efforts to model cardiovascular diseases in vitro (*Burridge et al., 2015*; *Lian et al., 2013*). While non-human model systems have provided significant insight into the developmental lineages that contribute to cardiac development, the inaccessibility of early human embryonic tissue has significantly limited the creation of an in vivo reference atlas of human heart field development. Importantly, questions remain as to whether the human iPSC system can be used to efficiently generate cell types representing distinct chambers of the heart such as left/right ventricular cardiomyocytes, outflow tract cardiomyocytes, or atrioventricular canal cells. Currently, a major gap

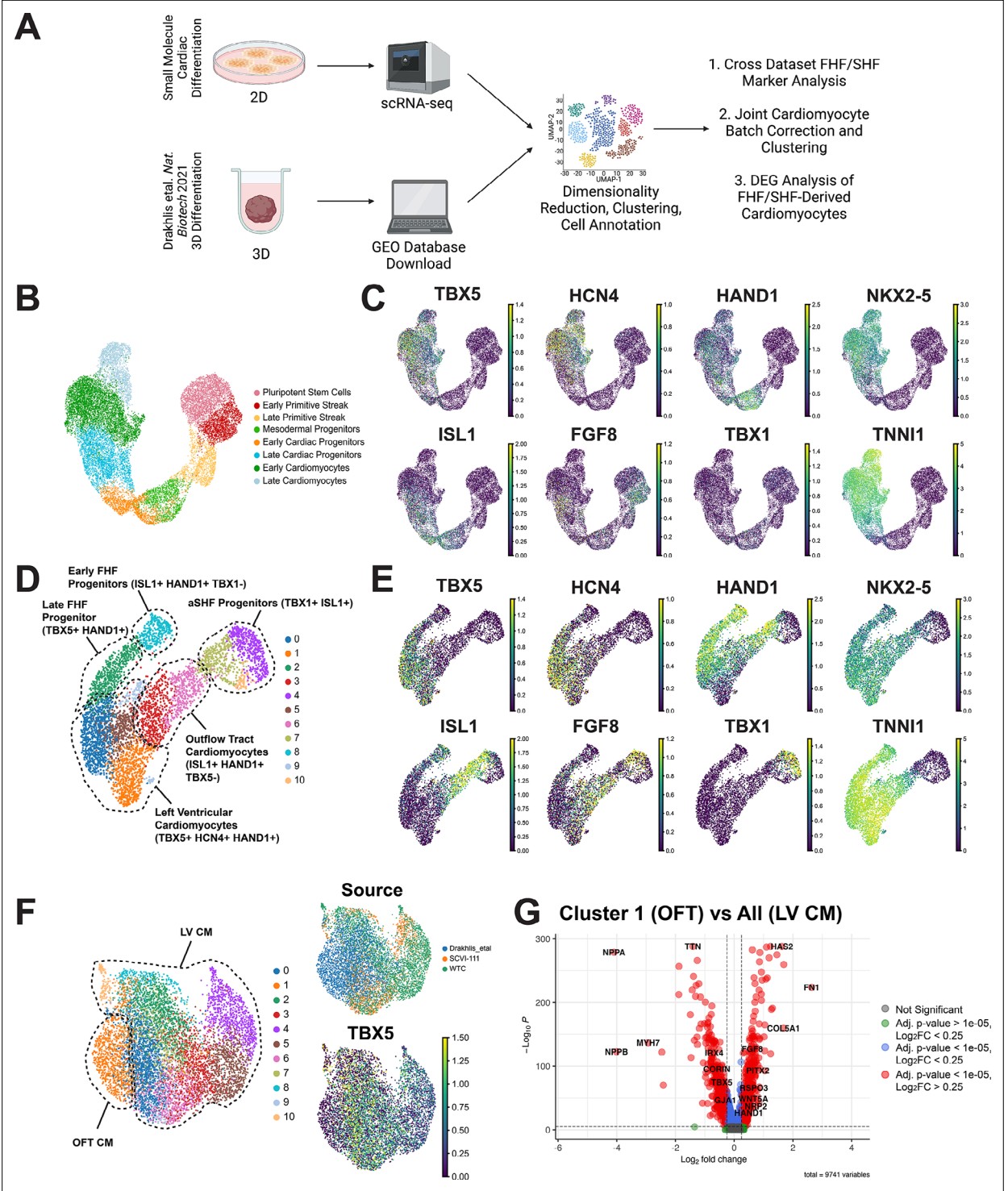

**Figure 6.** Comparison of 2D and 3D cardiac differentiation uncovers potential of organoid system for second heart field (SHF) generation. (**A**) Schematic of comparison strategy of single-cell RNA sequencing (scRNA-seq) data from 2D human-induced pluripotent stem cell (hiPSC) cardiac differentiation generated using our 2D protocol and a previously published organoid protocol. (**B**) UMAP plot of the myocardial lineage identified during 2D hiPSC cardiac differentiation. (**C**) Feature plots of first heart field (FHF) (TBX5, HCN4, HAND1), pan-cardiac (ISL1, NKX2-5, TNNI1), and SHF (FGF8, TBX1) during 2D hiPSC differentiation. (**D**) UMAP plot of Drakhlis et al., cardiac lineage cells with annotation of unsupervised clusters. (**E**) Feature plots of FHF (TBX5, HCN4, HAND1), pan-cardiac (ISL1, NKX2-5, TNNI1), and SHF (FGF8, TBX1) during organoid differentiation. (**F**) Co-clustering of 2D and 3D cardiomyocytes with annotation of source dataset (right-top) and expression of T-box transcription factor (TBX5) (right-bottom). (**G**) Volcano plot for top differentially expressed genes between OFT identified cluster and the remainder of identified cardiomyocytes identified as left ventricular.

*Figure 6 continued on next page*

*Figure 6 continued*

The online version of this article includes the following source data for figure 6:

**Source data 1.** Differential gene expression analysis between putative OFT and LV cardiomyocyte clusters in joint 2D and 3D hiPSC-CM Data.

exists in the ability to model the development of specific structures of the human heart in part due to the lack of genetic tools to mark distinct cell lineage in vitro. Importantly, the ability to identify chamber-specific cardiomyocyte is vitally important to modeling the early developmental mechanisms that give rise to structural congenital heart defects (*Doyle et al., 2015*; *Reller et al., 2008*; *van der Linde et al., 2011*).

In this paper, we sought to build a novel genetic lineage tracing tool to elucidate the identities of cardiomyocytes generated using a well-cited and standard differentiation protocol. We successfully implemented a *TBX5* lineage tracing scheme into the hiPSC system by targeting a highly sensitive Cre recombinase to the 3' end of the endogenous *TBX5* locus. Moreover, by including an *MYL2*-TdTomato direct reporter in our hiPSC lines, we evaluated the percentage of ventricular cardiomyocytes that were descended from *TBX5*-expressing precursors. By implementing a lineage tracing method, our reporter system provides several advantages for studying the descendants of FHF progenitor cell types compared with previous approaches. While previous studies have shown the isolation of TBX5-positive cell types using direct reporter schemes (*Zhang et al., 2019*), a major advantage of a lineage tracing approach is the permanent and robust labeling of descendants from progenitor populations or cardiomyocyte populations that express *TBX5*, thus allowing for the evaluation of cell type identity at later stages of differentiation when TBX5 expression is downregulated. Moreover, the combination of our lineage tracing and a ventricular reporter system allowed us to evaluate the proportion of definitive ventricular cardiomyocytes from the TBX5-lineage during iPSC differentiation that generates atrial cardiomyocytes as well.

Surprisingly, our data indicate that in two distinct hiPSC lines, TBX5-lineage-positive cardiomyocytes represent more than 95% of all cardiomyocytes generated. Furthermore, our scRNA-seq time course data further revealed the emergence of a core cardiac progenitor differentiation trajectory that displayed a gradual upregulation of FHF markers and no expression of known SHF genes. The unexpected finding of FHF predominance raises important questions on whether the widely used small molecule WNT modulation protocol used for the generation of hiPSC cardiomyocytes is biased towards the generation of FHF lineage cardiomyocytes. An additional finding from our scRNA-seq analysis that was quite intriguing was the bifurcation of cardiac progenitors into epicardial and myocardial cells. Importantly, the epicardial lineage arose from a cell cluster that exhibited high expression of *HAND1* representing early FHF progenitor cells. Recently, two groups have published lineage tracing results in mice that suggest a subset of FHF progenitor cells exhibit contributions to both the proepicardial and myocardial lineages (*Tyser et al., 2020*; *Zhang et al., 2021*). Consistent with these studies, our data provides evidence of this bifurcation in vitro and demonstrates the high expression of left ventricular markers *HAND1, TBX5*, and *HCN4* along the myocardial lineage. The high expression of *HAND1* during the earliest stages of mesoderm differentiation further suggests that even at the earliest stages of mesoderm specification the progenitors in our hiPSC differentiations were already bound for an FHF fate given the known restriction of *HAND1* to give rise to FHF-derived cell types during early embryonic development (*Barnes et al., 2010*; *Vincentz et al., 2017*). This observation is consistent with a growing body of literature suggesting that the bifurcation of the first and second heart fields occurs during the earliest patterning of mesoderm during its emergence from the primitive streak (*Lescroart et al., 2014*).

Having established the FHF identity of our cardiac differentiation conducted in 2D, we used our dataset as a reference point for determining whether a recently published cardiac organoid protocol could give rise to a greater diversity of cardiac progenitor cell types. While multiple 3D protocols have recently been published, the study by Drakhlis et al., is the only protocol to date to demonstrate the co-emergence of anterior endoderm cell types that are closely related to the emergence of an anterior cell types such as the anterior second heart field during embryonic development. Interestingly, our comparison revealed the presence of *bona fide* second heart field progenitors within the organoids generated in the Drakhlis et al., protocol. Importantly, we were able to combine cardiomyocytes generated from our study with those from the Drakhlis et al., 3D protocol and reveal the identity of a true OFT cell type present in the Drakhlis et al., dataset (*Figure 6F*). This analysis thus reveals the

promising application of 3D differentiation protocols for generating a greater diversity of cardiac cell types.

Overall, here we provide a novel reporter system that allows for the identification of left ventricular cardiomyocytes during the course of hiPSC differentiation. By allowing for the permanent labeling of *TBX5* descended cell types, we envision our system being used to conduct more complex studies to study chamber-specific cardiomyocytes in the context of congenital heart diseases as well as for the development of novel hiPSC differentiation protocols for generating both left and right ventricular cardiomyocytes. Moreover, by generating a scRNA-seq dataset profiling multiple consecutive days of hiPSC cardiac differentiation we provide here a reference atlas of the differentiation events that occur during human in vitro cardiac development. Together, our study provides extensive evidence of the identification of the FHF lineage in a human system and reveals the early bifurcation of this lineage into an epicardial and myocardial lineage.

## Materials and methods
### Cell lines
hiPSC lines used in this study were obtained from the Stanford Cardiovascular Institute Biobank (SCVI-111, Sendai virus reprogrammed peripheral blood mononuclear cells, healthy male with normal karyotype, 46, XY). The WTC-11 (reprogrammed from healthy males with normal karyotype, 46, XY) hiPSC line was provided by Bruce Conklin's laboratory at the University of California, San Francisco, and has been deposited into the Coriell Institute for Medical Research under identifier GM25256. For SCVI-111, G-banding karyotyping was conducted and cell line identity was confirmed by short tandem repeat analysis of the cell line and donor PBMCs. For the WTC-11 line, G-banding karyotyping was conducted and cell line identity was confirmed by short tandem repeat analysis of the cell line to donor fibroblasts. All cell lines tested negative for mycoplasma. Studies involved human iPSCs approved under protocol #460 of the Stanford Stem Cell Research Oversight (SCRO) committee.

### Cardiac differentiation
hiPSCs were maintained in DMEM/F12 (Corning Cat. 10–092 CM) supplemented with essential eight (E8) (henceforth referred to as E8 media) that is prepared in-house as previously described (*Burridge et al., 2015*) and cultured on growth factor reduced Matrigel (Corning Cat. 356321) coated plates at a 1:300 dilution. Upon reaching 75–80% confluency, hiPSCs were passaged using 0.5 mM EDTA in PBS for 8 min at 37 °C. Passaging was conducted with gentile dissociation of cell clusters and plated in E8 media supplemented 10 μM ROCK inhibitor (Selleckchem Cat. S1049). Passaging was performed using a 1:12 splitting ratio to achieve approximately 10,000 cells per cm$^2$. 24 hr after passaging media was changed to E8 media. Daily media changes were conducted until cells reached 90–95% confluency at which point media was changed to RPMI-1640 (Corning Cat. 10–040-CV) containing 6 μM CHIR99021 (Selleckchem Cat. CT99021) and 2% B27 minus insulin supplement (Thermo Fisher Cat. A1895601). Two days after initial treatment with CHIR, media was changed to RPMI-1640 with 2% B27 minus insulin for 24 hr. Between days 3–5, media was changed to 2 μM C59 (Selleckchem Cat. S7037) in RPMI-1640 media with 2% B27 minus insulin. On day 5 of differentiation, media was changed for RPMI-1640 with 2% B27 minus insulin for 48 hr and was subsequently changed to RPMI-1640 with 2% B27 Plus Insulin (Thermo Fisher Cat. 17504044) for another 48 hrs. On day 9, cells underwent glucose deprivation for 48 hr to purify cardiomyocytes by changing media to RPMI-1640 minus glucose with 2% B27 Plus insulin. Cardiomyocytes were subsequently maintained in RPMI-1640 with glucose with 2% B27 Plus Insulin.

### Donor construct plasmids
Cre recombinase gene sequence was provided by Connie Cepko lab (Addgene plasmid # 13775) (Matsuda and Cepko, 2007). TurboGFP gene was obtained from the pMaxGFP plasmid obtained from the Lonza P3 Primary Cell 4D-Nucleofector X Kit L. Plasmids were constructed using the plasmid construct service by Genscript Biotech. All donor plasmids are freely available upon request.

## CRISPR/Cas9 genome editing

Genetic constructs were targeted to hiPSCs following the schematic presented in *Figure 1*. TurboGFP sequence was cloned from the pMaxGFP plasmid from Lonza P3 Primary Cell Nucleofection kit. Protocol for targeting of genetic constructs was followed as previously described (*Galdos et al., 2021*). Briefly, hiPSCs were dissociated into a single-cell suspension at 75% confluency using an Accutase-EDTA solution (Millipore Cat. SCR005) containing a 0.02% blebbistatin (Sigma-Aldrich Cat. B0560). Dissociation reaction was quenched using a solution of E8 media with 10 μM ROCK inhibitor an 0.02% blebbistatin. Cells were then pelleted by centrifugation at 200 g for 3 min. We subsequently conducted nucleofection of the dissociated cells using the Lonza P3 Primary Cell 4D-Nucleofector X Kit L. A transfection mix was prepared to contain the Lonza P3 Solution, Supplement, single guide RNA/Cas9 expressing plasmid (1 μg), and the donor template plasmid (3 μg). Single Guide RNA sequences are found in *Figure 1—source data 2*. Cells were electroporated using a Lonza 4D Nucleofector machine using protocol number 'CM150.' After electroporation, 1 mL of E8 media supplemented with 10 μM ROCK inhibitor was gently added to the cuvette containing the cells. Cells were allowed to rest for 10 min after which they were plated onto two wells of a six-well plate. 24 hr after plating, the media was changed to regular E8 media, and regular maintenance until cells reach 50% confluency. At the 50% confluency mark, cells were dissociated into single-cell suspensions and passaged onto six-well plates at 1000 cells per well of six-well. We subsequently maintained the transfected hiPSCs in E8 media supplemented with appropriate antibiotics for the selection of successfully targeted cells. Concentrations were as follows for the constructs targeted: TBX5-Cre (Hygromycin 150 μg/mL Thermo Fisher Cat. 10687010), CCR5-CLSL-TurboGFP (G418 150 μg/mL Sigma Cat. 4727878001), and MYL2-Tdtomato (Puromycin 0.2 μg/mL Sigma Cat. P8833). After 5 days of treatment with antibiotics, cells were switched to regular E8 to allow for colony expansion derived from single-cells plated. After 4 days of colony expansion, colonies were picked into 24-well plates containing E8 plus 10 μM ROCK inhibitor and were expanded for downstream DNA extraction using Qiagen DNeasy Kit and for cell freezing using Bambanker.

PCR validation of construct integration was conducted using the 'Inside-Outside' approach where one primer was designed outside of the homology arms of the donor template and one primer was designed insight of the construct to be integrated. PCRs were conducted using the GoTaq Master Mix (Promega Corporation Cat. M7122), and products were run on a 1% agarose gel in 1 X TAE Buffer. PCR primer sequences are found in *Figure 1—source data 3*.

## Immunofluorescence staining

Immunocytochemistry was carried out for hiPSCs after genome editing and clonal selection for pluripotency marker OCT4 (Thermo Fisher Cat. MA1-104), Nanog (Thermo Fisher Cat. MA1-017), and Tra-1-8-1 (Stem Cell Technologies Cat. Tra-1–81). Cells were fixed in 4% Paraformaldehyde solution in PBS for 15 min. Cells were subsequently, washed three times for 5 min in 1 X PBS. The PBS was gently aspirated from cells and cells were incubated in a blocking solution composed of 1% Bovine Serum Albumin (Sigma Cat. A7906), 0.1% Triton-X100 (Sigma Cat. T8787) in PBS, and 1% Goat serum (Sigma Cat. 9023), for 1 hr at room temperature. After blocking, mouse anti-OCT4 (2 μg/mL), mouse anti-Nanog (1:100 dilution), or mouse anti-Tra-1-8-1 (5 μg/mL) antibodies were diluted in the blocking solution. Cells were incubated in primary antibody solution overnight at 4°C. The next day, the primary antibody was aspirated, and cells were washed three times in wash buffer (0.1% Tween-20 in PBS) for 5 min each. Cells were then rinsed in 1 X PBS and then incubated in a secondary antibody solution consisting of Goat anti-mouse Alexa Fluor 647 (Thermo Fisher Cat. A32728) secondary antibody at a 1:500 dilution and NucBlue DAPI (Thermo Fisher Cat. R37606) stain in blocking solution for 1 hr at room temperature and protected from light. Next, the secondary antibody solution was aspirated and cells were washed three times in wash buffer and rinsed once in 1 X PBS. Finally, chamber slide cover slips were mounted using Diamond Anti-Fade mounting (Thermo Fisher Cat. P36961) media and were subsequently imaged.

For cardiac troponin (cTnT) staining, we followed the same staining protocol as described above for pluripotency markers, however, we used mouse anti-cardiac troponin T antibody (Thermo Fisher Cat. MA5-12960) at a 1:300 dilution during the primary antibody incubation. Goat anti-mouse Alexa Fluor 647 (Thermo Fisher Cat. A32728) antibody was used at a 1:500 dilution along with a NucBlue DAPI counterstain.

## Flow cytometry

A Beckman Coulter CytoFLEX flow cytometer was used for high throughput analysis of TNNT2, TurboGFP, and TdTomato expression across time of hiPSC cardiac differentiation. On the day of time-point collection, cells were dissociated into single-cell suspensions by incubating in 10 X TrypLE Select (Thermo Fisher Cat. A1217701) for 5 min at 37°C. For later-stage cardiomyocytes (day 15 and onwards) incubation time was extended to 10 min to achieve single-cell dissociation. Cells were subsequently pelleted by centrifugation at 200 g for 5 min. Cell pellets were resuspended in 4% PFA for 10 min and were rinsed with a 5% KnockOut Serum Replacement (Thermo Fisher Cat. 10828028) solution in 1 x PBS. Cells were permeabilized in a 0.5% Saponin (Sigma Cat. S7900) solution containing 5% FBS in 1 X PBS (hereafter referred to as Saponin Solution). After permeabilization cells were incubated for 45 min in a monoclonal mouse anti-Troponin primary antibody (Thermo Fisher Cat. MA5-12960) at a 1:200 dilution in 0.5% saponin solution. Cells were rinsed twice in saponin solution and then incubated in secondary antibody AlexaFluor 647 goat anti-mouse (Thermo Fisher Cat. A32728) at a 1:1000 dilution in 0.5% saponin solution. Cells were subsequently rinsed in 1 x PBS twice and analyzed using CytoFLEX flow cytometer.

For sorting of TurboGFP+ and TurboGFP- day 15 cardiomyocytes, independent biological replicates consisting of independent differentiations were sorted using a FACSAria Fusion. Cells were dissociated using TRYPLE Select (Thermo Fisher Cat. A1217701) incubation for 5 min to ensure cells were fully dissociated. After single-cell suspensions were obtained, TRYPLE Select was neutralized with an equal volume of replating media consisting of 10% Knockout Serum Replacement, 2% B27 Plus Insulin Supplement, and RPMI-1640 with glucose. Cells were centrifuged at 200 g for 5 min and pellets were resuspended in 1 mL of replating media. Sorting was done into 10 mL falcon tubes containing replating media. After sorting, cells were centrifuged at 200 g for 5 min and lysed for RNA collection using Trizol Reagent (Thermo Fisher Cat. 15596026). RNA extraction was done using the Zymo DIRECT-Zol extraction kit (Zymo Research Cat. R2052) per manufacturer's protocol. Quantitative RT-PCR was done as described below.

## RNA extraction and quantitative RT-PCR

Cells were collected for RNA extraction by dissociation with 10 X TrypLE Select and pelleted as described in the flow cytometry section. Cell pellets were then resuspended in 300 µL of TRIZOL reagent (Thermo Fisher Cat. 15596018) at room temperature for 3 min. After complete resuspension of the cells, RNA was extracted using the Zymo DIRECT-Zol extraction kit (Zymo Research Cat. R2052) per the manufacturer's protocol. Purified RNA was reverse transcribed into cDNA using the High-Capacity RNA-to-cDNA kit (Thermo Fisher Cat. 4387406). Quantitative PCR was subsequently run on a Biorad qPCR 384-well machine using the Biorad SYBR qPCR master mix. RT-qPCR primer sequences are found in *Figure 3—source data 1*.

## Sample preparation for multiplexed scRNA-seq

To prepare cells for scRNA-seq, we dissociated cells at desired timepoints by incubating them in 10 X TRYPLE Select for 5 min at 37°C. Cells were gently dissociated by repeated pipetting. For later time-points (Days 15 and 30), we extended the TRYPLE Select (Thermo Fisher Cat. A1217701) incubation by 5 min to ensure cells were fully dissociated. After single-cell suspensions were obtained, TRYPLE Select was neutralized with an equal volume of replating media consisting of 10% Knockout Serum Replacement, 2% B27 Plus Insulin Supplement, and RPMI-1640 with glucose. Cells were centrifuged at 200 g for 5 min to obtain cell pellets. For days 0–6, we resuspended cells in BamBanker freezing medium (GC LTEC Cat. 302–14681) and control rate freezed vials of cells at each timepoint collected to obtain 2 million cells per freezing vial. For days 7 onwards, we froze down cells in a cardiomyocyte freezing medium consisting of 90% Knockout Serum Replacement and 10% cell culture grade DMSO (Sigma Cat. D2650).

For running the scRNA-seq experiment, we thawed the desired timepoints for each experimental run by thawing vials of cells at 37 °C and adding replating media dropwise to each vial. Cells were then centrifuged at 200 g for 5 min to obtain cell pellets. We conducted two washes in sterile 1 X DPBS for each sample to wash away any remaining Knockout Serum that could interfere with the chemically modified-oligonucleotide (CMO) staining. Using the 10 X Genomics CellPlex kit, we added 100 µL of CMO to each cell pellet and resuspended the cells to allow the lipid-conjugated oligonucleotides to

bind to each of our samples. Each CellPlex CMO consists of a unique barcode that is used for identifying individual samples that are run within a single channel of a 10 X Genomics ship. After incubating for 5 min, 1000 µL of sterile 1 X DPBS was added to each sample, and all samples were centrifuged for 5 min at 200 g. Cells were transferred to 5 mL Eppendorf which allowed for two more 1 x DPBS washes with a total of 5 mL DPBS for each sample. Washing of unbound CMO was critical to ensuring minimal cross-contamination of CMOs when combining samples. After washing, we proceeded to follow the 10 X Genomics 3′ 3.1 with CellPlex protocol for cell capture and combined samples as listed in *Figure 4—source data 4*. We aimed to capture a total of 30,000 cells per well of a 10 X Genomics GEM Chip. After cell capture, we proceeded with preparing gene expression libraries and CellPlex libraries by following the 10 X Genomic manufacturer's protocol. Libraries were sequenced using an Illumina NovaSeq 6000 with S4 v1.5 flowcell reagents. We sequenced the gene expression libraries at a depth of 25,000 paired-end reads per cell and the CellPlex libraries at 5000 paired-end reads per cell. Base calling during sequencing was performed using Real-Time Analysis Version 3 software.

## Bioinformatic analysis of hiPSC scRNA-seq time course

Raw FASTQ files were obtained for gene expression and CellPlex libraries and were aligned using CellRanger-6.0.0 using the count function. We aligned the gene expression libraries to the prebuilt GRCh38 Human genome reference provided by 10 X Genomics at: https://support.10xgenomics.com/single-cell-gene-expression/software/downloads/latest. We aligned the CellPlex libraries using a list of CMO barcodes as a reference as found in *Figure 4—source data 4*. After alignment, we obtained gene-by-cell expression matrices containing individual counts for each gene detected for each individual cell captured. We also obtained matrices containing the counts for each CMO detected per cell.

Gene expression matrices for each single-cell run were corrected for ambient RNA contamination using the SoupX package v1.5.2 (*Young and Behjati, 2020*), which detects levels of ambient RNA contamination using empty droplets processed during the single-cell capture. Following ambient RNA correction, we then imported the CMO expression matrix into the Seurat R package version 4.1.0 and conducted log-ratio normalization using the NormalizeData function. To demultiplex each sample according to the CMO used for labeling, we ran the HTODemux function using default parameters. This function assigns cell labels according to the amount of CMO counts detected per cell. It also identifies cells that can be classified as singlets, doublets, or negative (did not stain for any of the CMO labels).

After sample demultiplexing, we removed doublet and negative cells from the dataset as part of our quality control pipeline. We then calculated the percent of all RNA counts belonging to ribosomal and mitochondrial genes. To further remove low-quality, dead, or doublet cells, we calculated the median percentage of mitochondrial and ribosomal RNA counts detected as well as the total RNA counts per cell and genes detected per cell. We calculated an upper and lower cutoff for the elimination of low-quality cells by calculating the threshold at three times the median absolute deviation above and below the median value of each of these quality control metrics. Cells above and below these cutoffs were eliminated and the cells passing quality control were used for subsequent analyses.

After quality control analyses, we proceeded to merge data from all three single-cell runs conducted. We normalized RNA counts per cell using the NormalizeData function in Seurat and using default parameters. We then proceeded to integrate the three single-cell runs by using the mutual nearest neighbor batch correction algorithm and using the individual as the batch correction variable (*Haghverdi et al., 2018*). This was done in order to correct for the technical variation that occurs from running single-cell samples in individual capture runs. To conduct the batch correction, we first identified a common set of integration features across the three runs integrated by running the SelectIntegrationFeatures function in Seurat. After identifying highly variable features commonly found between the datasets, we further filtered these features by removing features associated with the cell cycle in order to remove the effect of the cell cycle from downstream analyses and clustering. We then ran the RunFastMNN function from the SeuratWrappers version 0.3.0 package. After integration, we then proceeded with constructing the nearest neighbors graph by using the FindNeighbors function and used 15 principal components that were identified using the ElbowPlot method in Seurat. We then proceeded to conduct non-linear dimensionality reduction by running the RunUMAP function using 15 principal components. Lastly, we conducted unsupervised clustering using the

FindClusters function. We then conducted differential gene expression analysis to annotate clusters based on literature-reported cell markers.

## Lineage trajectory analysis using STREAM

To conduct lineage trajectory analysis of hiPSC cardiac differentiations, we imported our integrated Seurat object into an AnnData object using ScanPy v1.8.2 package in Python (*Wolf et al., 2018*). After obtaining an AnnData object, we then fit a principal graph to the UMAP plot calculated using Seurat using the seed_elastic_principal_graph and elastic_principal_graph functions in STREAM version 1.1. In addition to fitting the principal graphs, we also calculated STREAM pseudotime and ordered cells along this pseudotime using the principal graph calculation functions. After calculating pseudotime, we projected cells across distinct differentiation trajectories using the plot_flat_tree function and also plotted cells in a subway plot using plot_stream_sc. These functions allow for the visualization of cellular differentiation along distinct differentiation paths identified by the STREAM algorithm and for downstream identification of gene markers that are expressed during each trajectory identified. After obtaining distinct differentiation trajectory branches across pseudotime, we calculated the top differentially expressed genes across each major branch identified by STREAM by using the detect_transition_markers function.

## Comparison of hiPSC and murine cardiac development scRNA-seq

To compare our hiPSC differentiation scRNA-seq data with murine data, we downloaded data from the GEO database from three previously published datasets. We extracted cells of interest from these datasets representing nascent mesoderm, heart field progenitors, left and right ventricular cardiomyocytes, outflow tract cardiomyocytes, and epicardial cells. In order to jointly plot these cells across multiple different datasets, we used the Fast Mutual Nearest Neighbor algorithm and integrated the cells for each dataset together. We then proceeded to construct the shared nearest neighbor graph and conduct dimensionality reduction using the RunUMAP function by using the corrected principal components derived using the FastMNN algorithm.

To analyze the development of FHF and aSHF differentiation lineages, respectively, we subsetted cells that are known from the literature to fall along each lineage. For example, for the FHF differentiation trajectory, we reclustered cells identified as FHF Progenitors, epicardial cells, and left ventricular cardiomyocytes. For the anterior second heart field, we clustered SHF progenitors along with right ventricular and outflow tract cardiomyocytes. After subsetting the cells, we reran the FastMNN algorithm to recalculate the corrected PCA space for the subsetted cells and continued with deriving UMAP plots. These plots were then used to calculate principal graphs using STREAM and to identify the differentiation lineages that arise during the progenitor's differentiation trajectories. STREAM then allowed for the plotting of STREAM plots along the differentiation trajectories identified, where we probed for the expression of well-established FHF and aSHF genes. These genes were also evaluated using the human iPSC cardiac differentiations using STREAM plots.

## scRNA-seq Comparison of 2D and previously published 3D hiPSC Cardiac Differentiation

To compare our 2D hiPSC cardiac differentiations to a previously published cardiac organoid protocol, we downloaded data from Drakhlis et al., from GSE150202 and subsetted the dataset for putative cardiomyocytes and cardiac progenitors. The Drakhlis et al., dataset was generated from two individual heart-forming organoids, therefore, to correct for the batch effect from these two independent organoids, we ran the FastMNN algorithm to conduct dimensionality reduction using the FindNeighbors and RunUMAP functions of Seurat. Moreover, we conducted unsupervised clustering of the Drakhlis et al., dataset where we identified major clusters of cardiac progenitors and cardiomyocytes. To compare between our hiPSC 2D data and the Drakhlis et al., data, we plotted the expression of multiple FHF and SHF markers using features plots. Moreover, we focused on the myocardial lineage on our hiPSC 2D differentiations to conduct a direct comparison of the cardiomyocyte differentiation lineages in the datasets.

To conduct unsupervised clustering of both the Drakhlis et al., cardiomyocytes and those from our 2D differentiations, we merged only the cardiomyocytes from the datasets and batch corrected using FastMNN. We subsequently conducted dimensionality reduction using FindNeighbors and

RunUMAP in Seurat and conducted unsupervised clustering of the cardiomyocyte populations. We then conducted differential gene expression analysis by using the FindMarkers function in Seurat and compared the expression of the putative outflow tract cluster with the rest of the cardiomyocyte clusters. We then plotted a volcano plot using the EnhancedVolcano package which allowed for the visualization of statistically significant upregulated and downregulated markers in the OFT cluster relative to the other cardiomyocyte populations. Adjusted p-values for the differential expression analysis were calculated using Bonferroni correction for multiple comparisons.

## Statistics

For studies conducted in this manuscript, biological replicates were defined as independently conducted differentiations where hiPSCs were independently plated and carried through our standard differentiation protocol. Samples were collected at distinct timepoints for the respective experiment. For quantitative PCR data, for each biological replicate, we ran each gene for each timepoint in technical duplicates, and the average cycle threshold value for each technical replicate was taken for downstream analysis using the delta-delta-Ct method.

Data presented in bar graphs for flow cytometry and RT-qPCR are presented as a mean ± standard error of the mean. Two-way ANOVA was conducted for statistical analysis of flow cytometry data with the first independent variable analyzed being time and the second variable being cell line. One-way ANOVA with Dunnet's multiple comparisons correction was conducted for statistical analysis of RT-qPCR data. For flow cytometry data, outliers were removed based on the percentage of cardiac troponin-expressing cells using the ROUT method for outlier detection (*Motulsky and Brown, 2006*).

For differential gene expression analysis of single-cell data, we ran FindMarkers or FindAllMarkers function in Seurat which allows for the evaluation of differentially expressed genes between cell populations of interest. For each gene evaluated, the log base 2 of the fold change between the population of interest and the comparator population was calculated along with the adjusted p-values based on Bonferroni correction using all the genes in the dataset.

To identify the differential expression of genes along distinct differentiation branches during the STREAM analysis, we conducted the leaf gene detection which involves calculating the average gene expression of genes along the leaves of the developmental trajectory. Detailed explanations for the statistical calculations conducted by the STREAM package to find differentially expressed genes along differentiation branches can be found in the original STREAM publication (*Chen et al., 2019*).

## Acknowledgements

We would like to thank all members of the Sean Wu laboratory for their input and feedback on the work presented in this manuscript. A special thanks to the members of the Stanford Stem Cell Institute Flow Cytometry Core facility for their training and experimental advice. We would also like to thank Sneha Venkatramen and Daniel Lee for their assistance in supply management in the laboratory. We thank Vittorio Sebastiano's laboratory for providing us access to the Stem Cell Core electroporation equipment for genome editing experiments.

This work was supported by the NIH/NIGMS grant 1RM1 GM131981-02, American Heart Association - Established Investigator Award, Hoffmann/Schroepfer Foundation, Additional Venture Foundation, Joan and Sanford I Weill Scholar Fund, and NIHLBI grant R01HL13483004 (S.M.W). This work was also supported by the NIH K08 Mentored Clinical Scientist Research Career Development Award (NHLBI) (1K08HL15378501) (W.R.G.) and NIH/NHLBI F30Hl149152 grant, the Dorothy Dee and Marjorie Boring Trust Award, and the Stanford Medical Scientist Training Program (F.X.G).

## Additional information

### Funding

| Funder | Grant reference number | Author |
| --- | --- | --- |
| NHLBI Division of Intramural Research | F30Hl149152 | Francisco X Galdos |

| Funder | Grant reference number | Author |
|---|---|---|
| NHLBI Division of Intramural Research | R01HL13483004 | Sean M Wu |
| NIH Office of the Director | 1RM1 GM131981-02 | Sean M Wu |
| NIH Office of the Director | NIH T32 GM007365 | Francisco X Galdos |
| NIH K08 Mentored Clinical Scientist Research Career Development Award (NHLBI) | 1K08HL15378501 | William Goodyer |
| NIH/NHLBI | F30Hl149152 | Francisco X Galdos |
| The Dorothy Dee and Marjorie Boring Trust Award | | Francisco X Galdos |
| Stanford Medical Scientist Training Program | | Francisco X Galdos |

The funders had no role in study design, data collection and interpretation, or the decision to submit the work for publication.

## Author contributions

Francisco X Galdos, Conceptualization, Resources, Data curation, Software, Formal analysis, Funding acquisition, Investigation, Methodology, Writing – original draft, Project administration, Writing – review and editing; Carissa Lee, Adrija Darsha, Data curation, Formal analysis, Investigation; Soah Lee, Conceptualization, Methodology, Writing – review and editing; Sharon Paige, Conceptualization, Methodology; William Goodyer, Conceptualization; Sidra Xu, Aimee Beck, Investigation; Tahmina Samad, Writing – review and editing; Gabriela V Escobar, Formal analysis, Investigation, Methodology; Rasmus O Bak, Conceptualization, Investigation, Methodology; Matthew H Porteus, Methodology; Sean M Wu, Conceptualization, Resources, Data curation, Formal analysis, Supervision, Funding acquisition, Investigation, Methodology, Writing – original draft, Project administration, Writing – review and editing

## Author ORCIDs

Francisco X Galdos http://orcid.org/0000-0002-7985-4521
Rasmus O Bak http://orcid.org/0000-0002-7383-0297
Matthew H Porteus http://orcid.org/0000-0002-3850-4648
Sean M Wu http://orcid.org/0000-0002-0000-3821

## Decision letter and Author response

Decision letter https://doi.org/10.7554/eLife.80075.sa1
Author response https://doi.org/10.7554/eLife.80075.sa2

---

# Additional files

## Supplementary files
• MDAR checklist

## Data availability

All raw data for single cell RNA-sequencing has been deposited in the GEO repository under accession number GSE202398. Accession numbers for publicly available data re-analyzed for this study can be found in Supplementary File 9. Standard code and functions used for single cell analysis are available at the following Github repositories: Seurat (https://github.com/satijalab/seurat/, (copy archived at swh:1:rev:763259d05991d40721dee99c9919ec6d4491d15e)), ScanPy (https://github.com/scverse/scanpy, (copy archived at swh:1:rev:9cab8cfa4033d3f47a36c7bb816b2c9fae-5cfdc6)), STREAM (https://github.com/pinellolab/STREAM, (copy archived at swh:1:rev:d20c-c1faea58df10c53ee72447a9443f4b6c8e03)), SoupX (https://github.com/constantAmateur/SoupX, (copy archived at swh:1:rev:8d89492306a7e82a79a3c0588b806d5127f2003c)), CellRanger (https://

support.10xgenomics.com/single-cell-gene-expression/software/pipelines/latest/using/tutorial_ov).

The following dataset was generated:

| Author(s) | Year | Dataset title | Dataset URL | Database and Identifier |
|---|---|---|---|---|
| Galdos FX, SM Wu | 2022 | Combined Lineage Tracing and scRNA-seq Reveals Unexpected First Heart Field Predominance of Human iPSC Differentiation | https://www.ncbi.nlm.nih.gov/geo/query/acc.cgi?acc=GSE202398 | NCBI Gene Expression Omnibus, GSE202398 |

The following previously published datasets were used:

| Author(s) | Year | Dataset title | Dataset URL | Database and Identifier |
|---|---|---|---|---|
| de Soysa TY, Gifford CA, Srivastava D | 2019 | Single-cell analysis of cardiogenesis reveals basis for organ level developmental defects | https://www.ncbi.nlm.nih.gov/geo/query/acc.cgi?acc=GSE126128 | NCBI Gene Expression Omnibus, GSE126128 |
| Hill MC | 2019 | A Cellular Atlas of Pitx2-Dependent Cardiac Development | https://www.ncbi.nlm.nih.gov/geo/query/acc.cgi=GSE131181 | NCBI Gene Expression Omnibus, GSE131181 |
| Drakhlis L, Biswanath S, Farr C, Lupanow V, Teske J, Ritzenhoff K, Franke A, Manstein F, Bolesani E, Kempf H, Liebscher S, Schenke-Layland K, Hegermann J, Nolte L, Meyer H, de la Roche J, Thiemann S, Wahl-Schott C, Martin U, Zweigerdt R | 2020 | Human heart-forming organoids recapitulate early heart and foregut development - single-cell RNA sequencing data | https://www.ncbi.nlm.nih.gov/geo/query/acc.cgi?acc=GSE150202 | NCBI Gene Expression Omnibus, GSE150202 |
| Pijuan-Sala B, Griffiths JA | 2019 | Timecourse single-cell RNAseq of whole mouse embryos harvested between days 6.5 and 8.5 of development | https://www.ebi.ac.uk/arrayexpress/experiments/E-MTAB-6967/ | ArrayExpress, E-MTAB-6967 |

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
