## [Editor Report]

This study presents elegant lineage tracing results demonstrating that first heart field (FHF) generates a dominance (>90%) of left ventricular cardiomyocytes in human iPSCs. The authors developed a TBX5/MYL2 reporter system in order to demonstrate this, and have supported their results utilizing single-cell RNA-sequencing with oligonucleotide-based sample multiplexing and this also provides a single-cell transcriptomic atlas of human iPSCs undergoing cardiac differentiation. These differentiation pathways have been extensively studied in non-human models but this is the first demonstration of FHF progenitors giving rise to left ventricular cardiomyocytes in a human model system.

---

## [Decision Letter]

**Decision letter after peer review:**

Thank you for submitting your article "Combined Lineage Tracing and scRNA-seq Reveals Unexpected First Heart Field Predominance of Human iPSC Differentiation" for consideration by *eLife*. Your article has been reviewed by 3 peer reviewers, one of whom is a member of our Board of Reviewing Editors, and the evaluation has been overseen by Didier Stainier as the Senior Editor. The reviewers have opted to remain anonymous.

Essential revisions:

1) The question of beating has been raised by at least 2 reviewers below, please address these concerns and if feasible, provide a video of beating cardiomyocytes.

2) Please address Question 3 of Rev 1 to show additional myocyte maturation markers other than just TNNT2.

3) Please address Rev 2's question regarding TBX5 expression in SHF.

4) Please address Rev 2's question about ISL1/TBX1/FGF8 cells at Days 3-7 and whether they become cardiomyocytes?

5) RT-qPCR of Hand1, Isl1, etc. expression in sorted single- and double-positive populations would be useful for confirming that the lineage tracing technique is indeed labeling the correct cells.

*Reviewer #1 (Recommendations for the authors):*

This is a very interesting study where the authors applied a novel lineage tracing method to hiPSC-CM differentiation, which can advance our current understanding of human cardiac cell origin and development. Using hiPSCs, this study also bypasses the limitation of using human embryonic samples with the advantages of molecular profiling developing cardiac lineages in vitro. I have the following comments:

1. It is interesting to see a striking 90%+ cells differentiated into FHF lineage, which led to the question of Tbx5 specificity and possible transient Tbx5 expression in vivo. Meanwhile, Hand1, another FHF marker, showed a different induction pattern between two hiPSC lines (Figure 3B) as the WTC line showed a similar trend as Tbx5, but the SCVI line had a relatively lower induction followed by a downregulated tendency after peaked at day 7. How are Tbx5 and Hand1 labeling FHF different in the differentiation axis (in vitro and in vivo)?

2. I would be curious to see visual validation of differentiation maturity of the Tbx5+GFP+ population. How is the morphology of differentiated cells at different time points, beating/contractility, etc. in comparison to Tbx5- population. It would be a strongly supportive evidence to show Tbx5+ and Tbx5- lineage cells beating at clearly different ventricular vs. atrial contractility.

3. In addition to TNNT2, have the authors detected any myocyte maturation markers such as MYH6/MYH7, and Actinin to show myocyte fully differentiated into maturity?

*Reviewer #2 (Recommendations for the authors):*

A key concern of this study is the specificity of Tbx5 expression during early embryonic development. The authors use a non-inducible TBX5-Cre to label FHF cells. It is uncertain if there is any low level, transient, or ectopic Tbx5 expression in the very early SHF cardiac precursors or early common cardiac precursors (SUPPLEMENTARY FIGURE 4B). If there is, an explanation of the data of this manuscript would be different. Most genetic lineage studies in FHF with mouse models use Tbx5-CreERT2, and this animal line cannot tell if Tbx5 is transiently or ectopically in the very early SHF cells due to the tamoxifen induction limit.

The authors detected a significant number of ISL1/TBX1/FGF8 cells at Days 3-7 (FIGURE 3). What is the exact ratio of these cells?? Why don't these cells become cardiomyocytes? Or can only a small portion of these cells eventually become cardiomyocytes?

*Reviewer #3 (Recommendations for the authors):*

1. It would valuable to see evidence that CMs derived from the gene-edited lines and expressing both fluorophores still beat. A brief video of a double-positive cell beating or even just a line in the text would help make it clear that neither the gene-editing nor the exogenous protein expression is functionally impairing the CMs and thus potentially their expression patterns.

2. To demonstrate that the false positive rate for the TurboGFP, in particular, is low even after weeks in culture, it would be useful to show by the flow that hiPSCs after 30 days of culture in pluripotency conditions still have a small or nonexistent positive population.

3. RT-qPCR of Hand1, Isl1, etc. expression in sorted single- and double-positive populations would go a long way toward confirming that the lineage tracing technique is indeed labeling the correct cells.

4. The hiPSC colonies in Figure 1E look looser than I would expect. They can clearly differentiate into at least mesoderm and endoderm-like cells based on your results so it is not that worrying, but a supplemental figure with alkaline phosphatase staining or other pluripotency markers would help my confidence.

---

## [Author Response]

Essential revisions:1) The question of beating has been raised by at least 2 reviewers below, please address these concerns and if feasible, provide a video of beating cardiomyocytes.

We have addressed the question of beating from the reviewers (see below) and have provided videos of beating cardiomyocytes (Videos 1 and 2).

2) Please address Question 3 of Rev 1 to show additional myocyte maturation markers other than just TNNT2.

We have addressed the question of myocyte maturation by profiling the expression of MYH7 and MYL2 in addition to TNNT2.

3) Please address Rev 2's question regarding TBX5 expression in SHF.

We have addressed Rev 2’s question regarding TBX5 expression in SHF which was also raised in Rev 1’s question 1 regarding specificity of TBX5 in vivo (see below), namely we observed no transient/low level of TBX5 expression observed in mouse mesodermal progenitors or early cardiac progenitor cells from single cell RNAseq data (Galdos et al., Nature Comm 2022) consistent with lineage tracing data at from tamoxifen treatment at e6.5 and e7.5 of TBX5-CreERT2 mice (Devine et al., *eLife* 2014).

4) Please address Rev 2's question about ISL1/TBX1/FGF8 cells at Days 3-7 and whether they become cardiomyocytes?

We have addressed Rev 2’s question about ISL1/TBX1/FGF8 cells at Days 3-7 (see below) and explained why very few of these cells eventually become cardiomyocytes.

5) RT-qPCR of Hand1, Isl1, etc. expression in sorted single- and double-positive populations would be useful for confirming that the lineage tracing technique is indeed labeling the correct cells.

We have performed additional RT-qPCR experiments of Tbx5 and TurboGFP in sorted cells to confirm the lineage tracing results.

Reviewer #1 (Recommendations for the authors):This is a very interesting study where the authors applied a novel lineage tracing method to hiPSC-CM differentiation, which can advance our current understanding of human cardiac cell origin and development. Using hiPSCs, this study also bypasses the limitation of using human embryonic samples with the advantages of molecular profiling developing cardiac lineages in vitro. I have the following comments:1. It is interesting to see a striking 90%+ cells differentiated into FHF lineage, which led to the question of Tbx5 specificity and possible transient Tbx5 expression in vivo. Meanwhile, Hand1, another FHF marker, showed a different induction pattern between two hiPSC lines (Figure 3B) as the WTC line showed a similar trend as Tbx5, but the SCVI line had a relatively lower induction followed by a downregulated tendency after peaked at day 7. How are Tbx5 and Hand1 labeling FHF different in the differentiation axis (in vitro and in vivo)?

Thank you for this question regarding the in vivo differences in the expression of Tbx5 and Hand1. Previous studies have shown that HAND1 and TBX5 are expressed in the cardiac crescent in the FHF progenitor domain during murine development. Moreover, HAND1 has been shown to be expressed in early mesodermal precursors that consist of the first wave of cardiac progenitors to exit the primitive streak. In studies conducted by Lescroart et al. (https://pubmed.ncbi.nlm.nih.gov/25150979/), single cell RNA sequencing analysis of Mesp1+ mesodermal progenitors reveal that even prior to the formation of the cardiac crescent, early late Mesp1+ mesodermal progenitors already express FHF markers such as HAND1. Lineage tracing studies have also shown that HAND1 progeny labels both the left ventricle and some cells in the outflow tract (due to late expression of HAND1 in outflow tract precursors) (Barnes et al. *Dev Dyn* 2010: https://pubmed.ncbi.nlm.nih.gov/20882677/). More recently, studies have shown that a subpopulation of the FHF is HAND1+ and TBX5- and contributes to the expression of the epicardium and left ventricle which we similarly observe in our hiPSC scRNA-seq data where a HAND1+ early cardiac progenitor contributes to both the epicardial and myocardial cells (Tyser et al. *Science* 2020: https://pubmed.ncbi.nlm.nih.gov/33414188/; Zhang et al. *Circulation Research* 2021: https://pubmed.ncbi.nlm.nih.gov/34162224/).

On the other hand, TBX5 has been noted to be expressed in the FHF and is expressed specifically in the LV and atria during the looping and chamber morphogenesis stages between E8.25-E11.5 of murine development (Bruneau et al. *Development* 1999: https://pubmed.ncbi.nlm.nih.gov/10373308/). In addition lineage tracing of TBX5 by tamoxifen injection at e6.5 and e7.5 of TBX5-CreERT2;R26-LacZ and TBX5-CreERT2;R26-mTmG mouse development, respectively, show no labeling in the RV or outflow tract (Devine et al., *eLife* 2014)(https://pubmed.ncbi.nlm.nih.gov/25296024/), unlike HAND1 expression in outflow tract during late development (Barnes et al. *Dev Dyn* 2010: https://pubmed.ncbi.nlm.nih.gov/20882677/). In our in vitro scRNA-seq data, we observe that the earliest cardiac progenitors that arise from the mesoderm express HAND1, consistent with in vivo reports of early patterning of the mesoderm to the FHF fate (Lescroart et al. https://pubmed.ncbi.nlm.nih.gov/25150979/). Our scRNA-seq data further shows that an early HAND1+ cardiac progenitor population that differentiates into WT1+ epicardial-like cells and TBX5+ LV-cardiomyocytes.

With respect to our qPCR data in Figure 2B showing a slightly different pattern of HAND1 expression, we would like to clarify that while the SCVI line shows a downregulation of HAND1 at Day 15, 20, and 30, the fold change expression is 31432-fold, 82784-fold, and 41964-fold greater than Day 0, indicating that HAND1 continues to be highly expressed in the SCVI line at later timepoints during differentiation. We have added an additional marker on the Y axis of the graph to better demonstrate this on the plotting of the data. While we do not know exactly why the expression trend of HAND1 differs between the two hiPSC lines, we are encouraged by the fact that the qPCR data and the scRNA-seq data on HAND1 expression correlates nicely with one another and precedes the onset of TBX5 expression at Day 7. Thus, the dynamics of HAND1 expression and TBX5 expression appear consistent with those observed in mice in vivo.

2. I would be curious to see visual validation of differentiation maturity of the Tbx5+GFP+ population. How is the morphology of differentiated cells at different time points, beating/contractility, etc. in comparison to Tbx5- population. It would be a strongly supportive evidence to show Tbx5+ and Tbx5- lineage cells beating at clearly different ventricular vs. atrial contractility.

Using our scRNA-seq data of hiPSC cardiac differentiation we assessed the expression of well-known atrial (KCNA5, NR2F1, and VSNL1) and ventricular markers (IRX4, MYH7, and MYL2) showing that essentially no atrial gene expression is present in our hiPSC-CM population (Figure 4—figure supplement 7). This is consistent with our prior finding that our small molecule WNT modulation protocol generate entirely ventricular cardiomyocytes (Galdos et al., Nature Comm 2022) (https://pubmed.ncbi.nlm.nih.gov/36071107/). Our ventricular MYL2-tdTomato reporter also demonstrates a gradual upregulation of tdTomato expression from Day 15 onwards with hiPSC-CM maturation (Figure 2) leading to >95% of the tdTomato+ ventricular cardiomyoytes also being TurboGFP+. Given this observation, the rare GFP- cells are likely immature FHF progenitors that have yet reach the stage of TBX5 expression rather than being atrial or RV cardiomyocytes.

3. In addition to TNNT2, have the authors detected any myocyte maturation markers such as MYH6/MYH7, and Actinin to show myocyte fully differentiated into maturity?

We have added an additional time course analysis of MYH7 in addition to the ventricular-specific marker MYL2 (Figure 3). Previous literature has shown that MYL2 and MYH7 are both upregulated during the course of cardiomyocyte maturation. We show that both MYL2 and MYH7 progressively increase in their expression in our hiPSC-derived cardiomyocytes.

Reviewer #2 (Recommendations for the authors):A key concern of this study is the specificity of Tbx5 expression during early embryonic development. The authors use a non-inducible TBX5-Cre to label FHF cells. It is uncertain if there is any low level, transient, or ectopic Tbx5 expression in the very early SHF cardiac precursors or early common cardiac precursors (SUPPLEMENTARY FIGURE 4B). If there is, an explanation of the data of this manuscript would be different. Most genetic lineage studies in FHF with mouse models use Tbx5-CreERT2, and this animal line cannot tell if Tbx5 is transiently or ectopically in the very early SHF cells due to the tamoxifen induction limit.

We thank the reviewer for this very important question which is reminiscent of the history of ISL1 expression studies showing that ISL1 is expressed in both FHF and SHF in early development but SHF only in late development. By combining four large publicly available scRNAseq datasets of mouse development from epiblast and primitive streak to four-chambered heart at embryonic day 16.5 our recent study showed that TBX5 is expressed only in FHF and posterior SHF (i.e. atrial cardiomyocyte) progenitor cells at embryonic days 6.5 to 7.5 and LV and septal cardiomyocytes at embryonic days 9.25 to 10.5 (Galdos et al., Nature Comm 2022) (https://pubmed.ncbi.nlm.nih.gov/36071107/). Importantly, no transient TBX5 expression in mesodermal or anterior SHF (i.e. RV cardiomyocyte) progenitor cells was found. Consistent with this, lineage tracing studies using TBX5-CreERT2 pulsed with tamoxifen at e6.5 and at e7.5 also showed no expression in RV or outflow tract (Devine et al., *eLife* 2014)( https://pubmed.ncbi.nlm.nih.gov/25296024/). In this study we also observed no TBX5 expression prior to FHF progenitor cell in during hiPSC differentiation (Figure 5D) thus the only other TBX5-lineaged cardiomyocytes besides LV cardiomyocytes are atrial cardiomyocytes (i.e. FHF and posterior SHF descendants) which we did not observe in any of our scRNAseq data (Figure 4—figure supplement 6).

The authors detected a significant number of ISL1/TBX1/FGF8 cells at Days 3-7 (FIGURE 3). What is the exact ratio of these cells?? Why don't these cells become cardiomyocytes? Or can only a small portion of these cells eventually become cardiomyocytes?

Thank you for this excellent question. It is important to note that while ISL1, TBX1, and FGF8 expression are present at Days 3-7 of hiPSC differentiation by bulk RT-qPCR, this should not be interpreted as the presence of a single cell that is simultaneously expressing all three markers as would be expected for a SHF progenitor cell. To address the potential presence of a ISL1/TBX1/FGF8 triple positive cell using our scRNAseq data from hiPSC differentiations we plotted the expression of ISL1, TBX1, and FGF8 on an individual cell basis to determine whether any cell at any point during the entire developmental time-course demonstrate expression of all three genes (see Figure 4—figure supplement 6). Our scRNAseq data shows that during Days 3 to 7 of differentiation ISL1 is expressed in late mesoderm to early cardiac progenitor cells which is consistent with previous studies (Ma et al. *Dev Bio* 2008: https://pubmed.ncbi.nlm.nih.gov/18775691/). In addition, ISL1 is also expressed in endodermal cells on Day 3-7 which is consistent with the ISL1 expression noted in our qPCR data (Figure 3). Consistent with our qPCR data, the TBX1 expression in scRNA-seq data is generally quite low with only scattered expression among pluripotent cells, endodermal cells, few mesodermal cells, and cardiac and epicardial progenitors Figure 4—figure supplement 6 The expression of FGF8 in our scRNA-seq data overlaps with ISL1 in mesoderm and in cardiac progenitor cells but minimally overlaps with TBX1 expression. These double ISL1/FGF8 positive cells are likely FHF cardiac progenitors expressing low level of ISL1 and FGF8 rather than SHF cardiac progenitors since these cells also express TBX5 but not TBX1 (Figure 4—figure supplement 6). It is important to note that the expression level of ISL/TBX1/FGF8 are generally very low (less than 10 to 1000-fold) compare with the expression of HAND1 and TBX5 that range from 1000 to 100,000-fold greater than pluripotent stem cells at Day 0. We have added additional text to the manuscript clarifying this point in lines 227-230. Take all together, we believe the there are few if any triple positive ISL1/TBX1/FGF8 cells and the ISL1/FGF8 double positive cells are predominantly FHF progenitor cells that eventually become LV cardiomyocytes.

We also note that unlike what was observed in the 3D organoid platform (Figure 6E) where we could see a clear differentiation trajectory from TBX1+ SHF progenitor cells to OFT-like cells we do not observe this in our 2D monolayer differentiation (see Figure 6F).

Reviewer #3 (Recommendations for the authors):1. It would valuable to see evidence that CMs derived from the gene-edited lines and expressing both fluorophores still beat. A brief video of a double-positive cell beating or even just a line in the text would help make it clear that neither the gene-editing nor the exogenous protein expression is functionally impairing the CMs and thus potentially their expression patterns.

We agree with the reviewer's comment and have now included videos of beating cardiomyocytes generated from our two gene-edited lines as Videos 1 and 2.

2. To demonstrate that the false positive rate for the TurboGFP, in particular, is low even after weeks in culture, it would be useful to show by the flow that hiPSCs after 30 days of culture in pluripotency conditions still have a small or nonexistent positive population.

Thank you for pointing out this important point. We now show a 30 day culture in pluripotency conditions and observe <2.3% of the hiPSCs will spontaneously express TurboGFP in the SCVI-111 line and <0.95% in the WTC line indicating a low false positive rate. This data is presented in Figure 1—figure supplement 1.

3. RT-qPCR of Hand1, Isl1, etc. expression in sorted single- and double-positive populations would go a long way toward confirming that the lineage tracing technique is indeed labeling the correct cells.

As suggested by the reviewer, we have sorted TurboGFP+ and TurboGFP- cells at Day 15-17 to determine the specificity of our reporter system. As seen from our flow cytometry time course the entirety of the cardiomyocyte’s expression GFP by Day 30 of differentiation which is where we see the upward rise of tdTomato expression. As a result, we were not able to obtain sufficient number of cells that are both TurboGFP-negative and tdTomato-positive for analysis.

Among the TurboGFP-positive and negative cells that were isolated, we observed that the positive cells are enriched for Tbx5 expression in both reporter hiPSC lines which provides evidence for the specificity of the lineage tracing system.

4. The hiPSC colonies in Figure 1E look looser than I would expect. They can clearly differentiate into at least mesoderm and endoderm-like cells based on your results so it is not that worrying, but a supplemental figure with alkaline phosphatase staining or other pluripotency markers would help my confidence.

We now show additional staining for pluripotency markers Oct4, Nanog, and Tra-1-8-1 in Figure 1E at the colony stage.